# Optimistic tree search strategies for black-box combinatorial optimization

**Cedric Malherbe**[1], **Antoine Grosnit**[1], **Rasul Tutunov**[1], **Jun Wang**[1,2],
**Haitham Bou-Ammar**[1,2]
[1]Huawei Noah's Ark Lab, [2]University College London
`firstname.lastname@huawei.com`

## Abstract

The optimization of combinatorial black-box functions is pervasive in computer science and engineering. However, the combinatorial explosion of the search space and the lack of natural ordering pose significant challenges for the current techniques from both theoretical and practical perspectives. In this paper, we propose to introduce and analyze novel combinatorial black-box solvers that are based on the recent advances in tree search strategies and partitioning techniques. A first contribution is the analysis of an algorithm called Optimistic Lipschitz Tree Search (OLTS) which assumes the Lipschitz constant of the objective function to be known. We provide linear convergence rates for this algorithm which are shown to improve upon the logarithmic rates of the baselines under specific conditions. Then, an adaptive version of OLTS, called Optimistic Combinatorial Tree Search (OCTS), is introduced for a more realistic setup where we do not have any information on the Lipschitz constant of the function. Again, similar linear rates are shown to hold for OCTS. Finally, a numerical assessment is provided to illustrate the potential of tree searches with respect to state-of-the-art methods over typical benchmarks.

## 1 Introduction

Finding optima of combinatorial black-box systems is an old problem with ubiquitous applications including but not limited to hyperparameter tuning [29], object detection [49], radar engineering [41] or model sparsification [7]. When facing such problems, two routes can generally be distinguished: 1) model-based techniques such as Bayesian methods [6, 14, 19, 26, 39] or 2) heuristic-based methods such as genetic and evolutionary algorithms [4, 21]. However, there is consistent gap between those two types of approaches. While relying on Bayesian modeling, the optimization only focuses on functions that align with the model and, more importantly, the heavy cost of optimizing the inner model restricts their use to systems where we can only afford few hundred function evaluations. On the other hand, heuristics methods are generally too coarse and tend to exploit very few information on the objective function which can get them easily stuck in local optima [31]. Further, the theoretical analysis of these approaches are generally loose when predicting their performance and limited to specific functions [16, 18, 20].

In this work, we aim at filling this gap by proposing the first provable black-box solver for combinatorial functions that rely on minimal assumptions. More precisely, we design novel solvers that come with provable guarantees and are tailored to optimize functions with moderate or cheap-to-evaluate cost. To do so, we follow a novel route and propose to develop an approach that employs ideas from tree searches and partitioning techniques. Precisely, we build upon the works of DIRECT [30] and SOO [37] and show how to use optimistic tree searches on combinatorial spaces. However, the application of these techniques to combinatorial structures is limited by three major drawbacks related to the absence of well suited trees for combinatorial spaces and the use of continuous notions that are further detailed in the next section.

36th Conference on Neural Information Processing Systems (NeurIPS 2022).

Providing solutions to these problems, our contributions can be summarized as follows. First, we identify novel conditions to create combinatorial trees tailored for the optimization of combinatorial structures (Section 3). Second, we show how to use these trees and develop the first combinatorial black-box solver with provable guarantees that rely on optimistic tree searches (Sections 4 & 5). Third, from a theoretical perspective, we obtain the first finite-time linear convergence rates that drastically improve upon the logarithmic rates of baselines, and explain the fast convergence of the solvers (Theorems 4.3 & 5.3). Finally, we provide a numerical assessment that illustrates the potential of tree searches with regards to existing techniques (Section 6).

## 2 Problem setting & related work

**Setup.** We focus on the problem of maximizing a black-box function over the Boolean hypercube [6, 43]. We denote by $\mathcal{X} = \{0,1\}^d$ the binary input space of dimensionality $d \geq 1$ and by $f : \mathcal{X} \to \mathbb{R}$ a real-valued objective function we wish to optimize. Our goal is to find an optimum $x^* \in \mathcal{X}$ solving:

$$x^* \in \operatorname*{argmax}_{x \in \mathcal{X}} f(x). \tag{1}$$

It is important to note that the paper adopts the standard black-box perspective in that $f$ cannot be assumed to have desirable characteristics such as linearity, quadraticity or submodularity that may facilitate the search for $x^*$. In fact, it is only assumed that the values of the objective function $f(x)$ can be evaluated at any candidate solution $x \in \mathcal{X}$ through an oracle that generally corresponds to a numerical evaluation of the function.

**Preliminaries.** A first strategy to solve Problem (1) would be to systematically enumerate and evaluate all possible candidates in the search space. Since the cardinality $|\mathcal{X}| = e^{d \ln(2)}$ grows exponentially with $d$, fully exploring $\mathcal{X}$ quickly becomes impractical even for moderate dimensionalities $d$. As such, standard optimization strategies attempt to approximate the global optimum by only evaluating $n \ll |\mathcal{X}|$ candidate solutions. To carefully choose those $n$ points, effective strategies rely on sequential procedures that choose the next evaluation point $x_{t+1} \in \mathcal{X}$ depending on information collected so-far $(x_1, f(x_1)), \ldots, (x_t, f(x_t))$ with $t$ denoting the iteration count.

**Notations.** For any $(x, x') \in \{0,1\}^{d \times 2}$, we denote by $d_H(x, x') = \sum_{i=1 \ldots d} \mathbb{I}\{x_i \neq x'_i\}$ their Hamming distance where $\mathbb{I}\{\cdot\}$ is the indicator function. We also utilize $\mathrm{Lip}(k) = \{f : \{0,1\}^d \to \mathbb{R} : |f(x) - f(x')| \leq k \cdot d_H(x, x'), \ \forall (x, x') \in \{0,1\}^{d \times 2}\}$ to represent the set of $k$-Lipschitz functions. For any subset $\mathcal{A} \subseteq \{0,1\}^d$, we further define its diameter as $\mathrm{Diam}(\mathcal{A}) = \max_{(x,x') \in \mathcal{A}^2} d_H(x, x')$. Lastly, $\lfloor x \rfloor$ and $\lceil x \rceil$ denote the floor and ceil operations and $x \wedge y = \min(x, y)$.

### 2.1 Related solution strategies

**Bayesian optimization.** Bayesian optimization has emerged as a powerful technique for the optimization of expensive-to-evaluate functions [13, 19] and has recently been extended to combinatorial spaces [6, 39, 48]. It relies on surrogate models and Bayesian probabilities to carefully selects the next evaluation point $x_{t+1} \in \arg\max_{x \in \mathcal{X}} a_t(x)$ by solving an inner optimization problem where $a_t(\cdot)$ is an acquisition function that depends on the chosen model. However, solving this inner optimization problem (often treated as a black-box problem) can be computationally as hard as directly optimizing the black-box function $f$ when its evaluation cost is cheap, which restricts their use to expensive-to-evaluate systems where we can typically only afford few function evaluations. Thus, the novelty of our work is complementary to Bayesian optimization in the sense that (1) it develops novel algorithmic ideas that aim at optimizing functions with moderate or cheap evaluation cost and (2) it provides novel theoretical insights for the generic problem of optimizing combinatorial black-box functions which are model-free.

**Evolutionary algorithms.** Evolutionary algorithms use mechanisms inspired by biological evolution, such as reproduction, mutation, recombination, and selection to optimize black-box functions [44]. They are generally the first line of algorithms attempted when facing black-box problems due to their flexibility. However, when compared to other techniques they do not always identify the global optimum of the function, can get stuck in local optima and are not necessarily sample-efficient (see, e.g., [25] and Section 6). In terms of theory, most works focus on specific synthetic problems such as the OneMax or LeadingOnes functions [16, 18, 20]. To the best of our knowledge, no finite-time bounds or sample-complexity results are known for these strategies. Thus, with regards to

evolutionary algorithms, the novelty of our work is (1) to propose a novel and orthogonal approach to combinatorial black-box optimization that relies on tree searches and (2) to derive generic finite-time analyses of the algorithms which hold for a large class of functions.

**Optimistic tree search strategies.** Optimistic tree search strategies [37] refer to approaches that implement optimism in face of uncertainty. This principle originated in the multi-armed bandit literature [2] and was later extended to tree searches. Similarly, tree based searches have also been discussed and analyzed in different settings such as UCT [11] and UCB [2]. The seminal work in [36] then adapted the optimistic principle to black-box optimization and introduced notions to analyze the convergence of these strategies in *continuous* spaces. More precisely, they provided sample complexity results characterized by two coefficients: $d$ and $C$, where $d$ is the near-optimality dimension (defined therein) and $C$ is a corresponding constant. Precisely, they show that objective functions with near-optimality dimension zero enable exponentially decreasing rates of optimistic tree search on continuous spaces. This, of course, opposes the polynomial rates attained by grid-search-like algorithms, for example. Later, this work was extended to settings with noisy evaluations [8, 23, 46], robust considerations [1] and Brownian optimization [24]. Though successful in isolated instances, none of the aforementioned algorithms are equipped to handle combinatorial spaces due to difficulties steming from the finite nature of the problem and the lack of natural ordering in discrete spaces. Succinctly, we identify the following three obstacles: (1) Contrary to continuous settings, it is not possible to define an infinitely finer partition of the search space restricting algorithmic design; (2) We cannot adopt continuous measures (e.g., near-optimality dimension) which depend on arbitrarily small neighborhoods around the optima to define complexities of combinatorial functions; (3) It is difficult to rely on bounds of the (rather standard) form: $\max_{x \in \mathcal{X}} f(x) - \max_{i=1...n} f(x_i) \leq Cn^{-\alpha}$ with large constants $C$ and attempt to analyze their exponent $\alpha$. This is due to the fact that in combinatorial spaces $n$ is (generally) upper bounded by $|\mathcal{X}|$, the cardinality of the search space $\mathcal{X}$. Hence, the associated upper-bound $C|\mathcal{X}|^{-\alpha}$ is not necessarily informative under a finite $n$. In this work, we fill this gap and show how to use optimistic machinery in combinatorial spaces. First, we introduce novel tree representations using hierarchical partitioning of combinatorial spaces. We then make the observation that all combinatorial functions are Lipschitz with $k_{\min} = \max_{x \neq x' \in \mathcal{X}^2} |f(x) - f(x')|/d_H(x, x')$, which we utilize to obtain discrete upper bounds of the form $(d - l(n))$ with $l(n) \in \mathbb{N}$. Importantly, our bounds only depend on the intrinsic properties of the black-box function like the conditioning of Definition 4.2. As a byproduct, we demonstrate that our optimistic tree search techniques can achieve *linear decreasing rates* compared to the logarithmic one exhibited by "naïve" strategies.

## 3 Tree representation of the combinatorial space

To implement optimistic tree search strategies [37], we need a hierarchical partition of the combinatorial space. A key ingredient of our approach lies in a specific nested tree representation of the input space $\mathcal{X} = \{0, 1\}^d$. Precisely, we consider a fully-balanced binary tree, $T$, of size $2^{d+1} - 1$ and depth $d$. The nodes of tree $T$, denoted by node $(l, i)$, are indexed by the values of the level $l \in \{0, \ldots, d\}$ and the corresponding index $i \in \{0, \ldots, 2^l - 1\}$ within that level. Each node $(l, i)$ is associated with a candidate solution $x_{l,i} \in \mathcal{X}$ where the objective function $f$ may be evaluated. Whenever a node is not terminal (i.e., $l < d$), each node $(l, i)$ is associated with a left child node $(l + 1, 2i)$ and a right child node $(l + 1, 2i + 1)$ at level $l + 1$. Moreover, since there are $2^{d+1} - 1 > |\mathcal{X}|$ nodes in the tree, a first specificity we consider is imposing that the value of the left child node $x_{l+1,2i}$ is equal to its parent node $x_{l,i}$ for any node.

**Assumption 3.1.** For any level $l \in \{0, \ldots, d - 1\}$ and $i \in \{0, \ldots, 2^l - 1\}$, we have $x_{l+1,2i} = x_{l,i}$

Importantly, this condition will be used in our approach to perform only one function evaluation when traversing levels in the tree. Additionally, we define the set $\mathcal{X}_{l,i}$ of child candidates associated with a node $(l, i)$ as the union of the candidate solution at this node and all the candidate solutions associated to its children.

**Definition 3.2. (Child nodes).** For any node $(l, i) \in T$, we define the set $\mathcal{X}_{l,i} \subseteq \mathcal{X}$ of children candidates recursively as follows:

$$\mathcal{X}_{l,i} := \{x_{l,i}\} \cup \mathcal{X}_{l+1,2i} \cup \mathcal{X}_{l+1,2i+1}, \quad \forall l \in \{0, \ldots, d - 1\},$$

where by convention, $\mathcal{X}_{d,i} = \{x_{d,i}\}$ for all terminal nodes with $i \in \{0, \ldots 2^d - 1\}$.

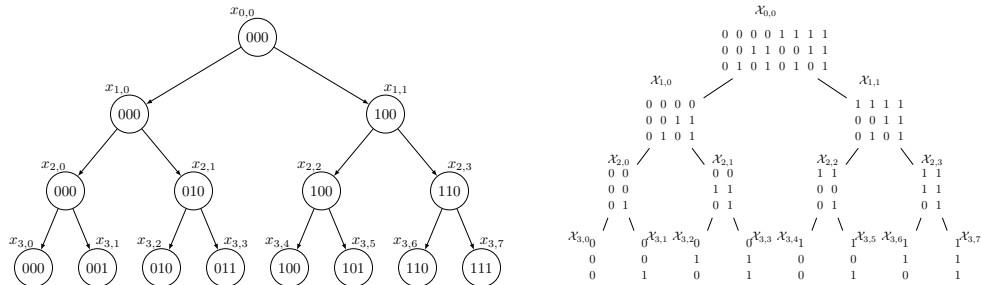

Figure 1: Left: example of the tree representation of the combinatorial space $\mathcal{X} = \{0, 1\}^d$ used when $d = 3$. The arrows represent the links between the nodes, and the nodes contain the value of $x_{l,i}$. Right: The set of children nodes $\mathcal{X}_{l,i}$ associated with each node$(l, i)$ of the tree forming a hierarchical partition of the search space satisfying the decreasing diameter property (Assumption 3.3).

It is important to note that the input space $\mathcal{X}$ can be partitioned into the $2^l$ disjoint sub-spaces $\mathcal{X}_{l,i}$ satisfying $\mathcal{X} = \cup_{i=1\ldots2^l} \mathcal{X}_{l,i}$ at any level $l \geq 0$. The last crucial property of the tree representations we consider is that of exhibiting a decreasing diameter in the following sense.

**Assumption 3.3. (Decreasing diameter).** We say that a tree $T$ satisfies the decreasing diameter property if, for any node$(l, i) \in T$, its set of children nodes $\mathcal{X}_{l,i}$ satisfies:

$$\text{Diam}(\mathcal{X}_{l,i}) := \max_{(x,x')\in\mathcal{X}_{l,i}^2} d_H(x, x') = d - l.$$

Informally, this condition states that if one picks any pair of candidates $(x_1, x_2) \in \mathcal{X}_{l,i}^2$ in a cluster $\mathcal{X}_{l,i}$ at any level $l$, we have that $d_H(x_1, x_2) \leq d - l$. This, in turn, ensures that the deeper we go in the tree, the closer the points bundle within a single cluster.

**Applicability of the assumptions.** Of course, the above conditions are not met by all trees. However, we can show that very natural representations of combinatorial spaces do, in fact, satisfy these assumptions. Actually, we identify that the trees which start at any root node and are split among $l$ components of either values of 0 or 1 do satisfy these conditions. Formally speaking, those trees (illustrated in Figure 1) can easily be constructed using the points $x_{l,i} = \text{Bin}_l(i) + \vec{0}_{d-l}$ where $\text{Bin}_l(i)$ denotes the binary representation of $i$ on $l$ bits (e.g., $\text{Bin}_3(0) = [0, 0, 0]$ and $\text{Bin}_3(2) = [0, 1, 0]$) and $\vec{0}_{d-l}$ denotes the vector filled with $d - l$ zeros. On top of this construction, it is also important to note that adding a vector or permuting the index of all the elements of these trees preserves these conditions (i.e., transformations $\pi(x_{l,i}) + c$ on all nodes of the tree for any $c \in \{0, 1\}^d$ and permutation $\pi : \{1, \ldots, d\} \to \{1, \ldots, d\}$).

**Comparison with continuous trees.** Finally, we point out that a comparison of the trees we propose to the trees used in continuous structures can be found in Appendix C.

## 4 Optimism in face of combinatorics: the known Lipschitz constant case

We now introduce a first algorithm to optimize combinatorial black-box functions given a priori knowledge of $f$ being Lipschitz, i.e., $f \in \text{Lip}(k)$ for a given constant $k \geq 0$. The aim of this section is to understand whether exploiting this information enables faster converging algorithms compared to exhaustive and random search. With those developed, we then extend our analysis to functions with unknown Lipschitz constants in Section 5.

### 4.1 The optimistic Lipschitz tree search (OLTS) algorithm

The OLTS algorithm (Algorithm 1) implements the optimistic principle [37] over combinatorial trees and aims at maximizing any function defined on the binary input space given a known Lipschitz constant $k \geq 0$. It starts by evaluating the function $f(x_{0,0})$ at the root node $(0, 0)$ of the tree and initiate the tree search by setting the root node as a starting point (line 2). At each iteration $t \geq 2$, the algorithm selects node $(l_t, i_t)$ with the highest value $f(x_{l_t,i_t}) + k\text{Diam}(\mathcal{X}_{l_t,i_t})$ from $\mathcal{T}_{t-1}$. Once $(l_t, i_t)$ is selected, OLTS evaluates the function on its children nodes adding them to the list of nodes of the current search. Note that the function is only evaluated on the right child as the value on its

---

**Algorithm 1** Optimistic Lipschitz Tree Search (OLTS)

---

**Require:** Tree representation $T$ of the search space $\{0,1\}^d$, Lipschitz constant $k \geq 0$, budget $n \geq 2$
1: Evaluate the objective function at the root node $f(x_{0,0})$
2: Initialize the tree search at the root node of the tree $\mathcal{T}_1 = \{(0,0)\}$
3: **for** $t = 2...n$ **do**
4:     Get the index of the node from the tree search with the highest upper confidence bound:

$$(l_t, i_t) \in \underset{(l,i)\in\mathcal{T}_{t-1}}{\arg\max} \ \{f(x_{l,i}) + k\text{Diam}(\mathcal{X}_{l,i})\}$$

5:     Evaluate the objective function on the right child of the selected node $f(x_{l_t+1,2i_t+1})$
6:     Remove the selected node from the tree search and add its child nodes if they are not terminal:

$$\mathcal{T}_t \leftarrow \mathcal{T}_{t-1}/\{(l_t, i_t)\} \cup \{(l_t + 1, 2i_t), (l_t + 1, 2i_t + 1)\}$$

7: **return** $x_{l_n,i_n}$ with $(l_n, i_n) \in \underset{\{l_t+1,2i_t+1\}_{t=2}^n\cup(0,0)}{\arg\max} \ f(x_{l,i})$

---

left child $f(x_{l_t+1,2i_t})$ is already known according to Assumption 3.1, whereby $x_{l_t+1,2i_t} = x_{l_t,i_t}$. Thus, the tree search is guided by two different quantities: (1) $f(x_{l,i})$ which represents the value of the function at a node and (2) $\text{Diam}(\mathcal{X}_{l,i})$ which indicates the size of the remaining children nodes below the $(l,i)^{th}$ node (Definition 3.2). Formally, this node selection rule can be explained by observing that since $f \in \text{Lip}(k)$, for any node $(l,i) \in \mathcal{T}_{t-1}$ in the search list, any of its child nodes satisfy: $f(x) \leq B_t(x) := f(x_{l,i}) + k\text{Diam}(\mathcal{X}_{l,i}), \ \forall x \in \mathcal{X}_{l,i}$. As such, OLTS explores the part of the tree $(i_t, l_t)$ that exhibits most promising values according to the previous upper bound. This selection process is often referred to as "optimism in the face of computational uncertainty" where the uncertainty comes from the potential function values. Although using a similar selection strategy to DOO [37], OLTS manifests three main differences: (1) once a node is selected, we switch one component of the node by evaluating the function on its right child and do not make several evaluations that depend on $d$, (2) we keep the left node value unchanged while only changing its level in the tree search when moving along the tree, (3) upon reaching level $d-1$, we do not add any nodes to the tree.

## 4.2 Theoretical guarantees for OLTS

Due to space constraints, we only present here the main results of our analysis and refer to Appendix D for more details. We start by casting generic convergence results for the algorithm.

**Proposition 4.1. (Convergence of the OLTS algorithm)** *Let $f \in Lip(k)$ be any objective function and let $I_l := \{nodes(l,i) \in T \text{ s.t. } f(x_{l,i}) + kDiam(\mathcal{X}_{l,i}) \geq f(x^*)\}$. Then, if $x_1,\ldots,x_n$[1] denote the set of evaluation points generated by OLTS tuned with constant $k$ after $n$ iterations over $f$ and the tree representation of Section 3, we have:*

$$\max_{x\in\mathcal{X}} f(x) - \max_{i=1...n} f(x_i) \leq k \cdot (d - l(n))$$

*where $l(n) := \min\{0 \leq L \leq d-1 : \sum_{l=0}^{L} |I_l| \geq n-1\}$.*

This result provides a generic finite-time bound on the approximation error that only depends on the cardinality of the set of potentially optimal nodes $I_l$ (further described in the Appendix D) and through the definition of $l(n)$. Here, $l(n)$ corresponds to the minimum depth at which the tree search will be after $n$ iterations. Of course, since the set $k$-Lipschitz functions contains very distinct functions, we expect the algorithm to display different behaviors captured here through the values of $|I_l|$. To gain a finer understanding, we thus bound this term using the following key complexity measure that describes the behavior of the function around its optimum $x^*$ through a local one-sided Lipschitz assumption.

**Definition 4.2. (Conditioning number).** Let $f : \mathcal{X} \rightarrow \mathbb{R}$ be any $k$-Lipschitz function. Then, if $f$ admits a unique global optimum $x^* \in \mathcal{X}$, we denote by $k^* \geq 0$ the largest value such that, for all $x \in \mathcal{X}$, we have:

$$f(x) \leq f(x^*) - k^* d_H(x, x^*)$$

---

[1] $x_1 = x_{0,0}$ and $x_t = x_{l_t+1,2i_t+1}$ for $t \geq 2$

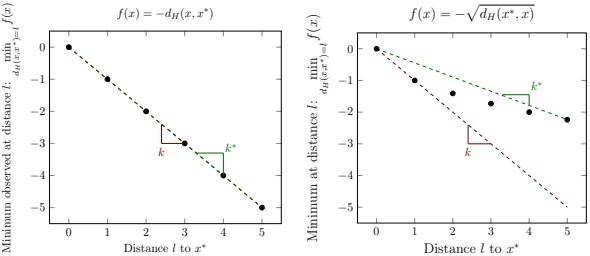

Figure 2: Computation of conditioning number on two functions $f(x) = -d_H(x, x^*)$ and $f(x) = -\sqrt{d_H(x, x^*)}$ with $x^* = [1]*5$ and $d = 5$. The black dots represent the minimum value of the function in terms of distance to the optimum $x^*$. The green and red lines represent the slopes of the Lipschitz constant $k$ and local smoothness coefficient $k^*$. With $k = 1$, for the left function we have $c = 1$ and $c = \sqrt{d}$ for the second function.

and we denote by $c = k/k^* \geq 1$ its conditioning number.

This conditioning number is closely related to measures used in continuous spaces [10, 34, 37]. However, it is interesting to note that since the input space $\mathcal{X}$ is discrete, the conditioning number is always defined [2] as opposed to the continuous space case. Moreover, Figure 2 provides an example of such computation. Equipped with Definition 4.2, we derive the following fast rate for OLTS.

**Theorem 4.3. (Fast convergence rates).** *Let $f \in Lip(k)$ be any combinatorial function with a unique maximiser with conditioning number $c = k/k^* < d - 1$. Then, if $x_1, \ldots, x_n$ denotes the series of evaluation points generated by OLTS tuned with constant $k \geq 0$ after $n$ iterations, there exists some $n_c \leq 2^{\lceil \frac{2c}{1+2c} d \rceil}$ and $C \leq 2^{\left(\frac{3c}{2c+1}\right) \frac{d}{2}}$, such that*

$$\max_{x \in \mathcal{X}} f(x) - \max_{i=1\ldots n} f(x_i) \leq k \times \begin{cases} d + 1 - \left\lceil \frac{\ln(n)}{\ln(2)} \right\rceil, & \text{if } n \leq n_c \\ d + 1 - \left\lceil \frac{\ln(n_c)}{\ln(2)} + \frac{n - n_c}{C} \right\rceil, & \text{if } n > n_c. \end{cases} \tag{2}$$

To gauge the importance of OLTS, we compare the convergence rate from Theorem 4.3 to those of exhaustive and random search of order $k(d - \ln(n)/\ln(2))$ that we derive in Appendix B. Our results indicate that we obtain improved linear rates over the logarithmic ones of such baselines and highlight two regimes of operation that depends on the value of $\log(n_c)$ which corresponds to the level after which the number of potential optima per level stops to explode and the algorithm will get a linear behavior. Namely, when $n < n_c$, we attain similar rates to exhaustive search. But upon passing this threshold $n > n_c$, the OLTS starts achieving linear rates with a constant $C$ by exploiting previous evaluations and focusing on promising areas. Hence, there is an exponential gain with regards to the convergence of naïve methods. To the best of our knowledge, it is the first result of this kind. Moreover, the conditioning number $c$ has both an impact on the length of the exploration phase $n_0$ and on the rate of convergence $C$. Thus, we deduce that to enjoy best convergence rates, the value of the Lipschitz constant $k \geq 0$ has to be chosen as small as possible.

## 5 Optimism in face of combinatorics: the unknown Lipschitz constant case

In this section, we introduce an extension of the previous algorithm to black-box settings where no knowledge of the Lipschitz constant is provided.

**Potentially optimal nodes.** Similar to the previous case, our algorithm will perform a search on the tree representation of Section 3. We thus require a technique to identify the set of potentially optimal nodes to guide the search. Since the Lipschitz constant is not known, we cannot get an upper bound indicating the potential values of the function as conducted in Section 4. To overcome this issue, we propose to define a novel set of potentially optimal nodes that do not explicitly depend on the Lipschitz constant of the objective function.

**Definition 5.1. (Potentially optimal nodes)** Let $\mathcal{T} = \{(l_1, i_1), \ldots, (l_t, i_t)\}$ be any set of index of the tree where we have an evaluation of the objective function $f(x_{l_i, i_i})$. Then, a node$(l, i)$ from the tree search $\mathcal{T}$ is said to be potentially optimal if there exists some Lipschitz constant $k > 0$ such that $f(x_{l,i}) + k\text{Diam}(\mathcal{X}_{l,i}) \geq f(x_{l',i'}) + k\text{Diam}(\mathcal{X}_{l',i'})$ for all $(l', i') \in \mathcal{T}$.

Informally, this definition states that a node is potentially optimal if there exists some Lipschitz constant such that the standard upper confidence bound evaluated on this node will be the highest

---

[2]with $k^* := \max_{1 \leq l \leq d}(f(x^*) - \min_{x \in \mathcal{X}: d_H(x, x^*) = l} f(x))/l$

---

**Algorithm 2** Optimistic Combinatorial Tree Search (OCTS)

---

**Require:** Tree representation $T$ of $\{0,1\}^d$, budget $n \geq 2$

1: Evaluate $f(x_{0,0})$, $t \leftarrow 1$
2: $\mathcal{T}_1 \leftarrow \{(0,0)\}$
3: **while** $t < n$ **do**
4:     Get the potentially optimal nodes of the tree search (Algorithm 3 in the Appendix): $\mathcal{T}_t^* \leftarrow$ OptimalNodes$(\mathcal{T}_t)$
5:     **for** node$(l^*, i^*)$ in $\mathcal{T}_t^*$ **do**
6:         Evaluate the function $f(x_{l^*+1, 2i^*+1})$ on the right child of the node$(l^*, i^*)$,   $t \leftarrow t+1$
7:         Remove the node$(l^*, i^*)$ from the tree search and add its children if they exist:

$$\mathcal{T}_t \leftarrow \mathcal{T}_{t-1} \cup \{(l^*+1, 2i^*), (l^*+1, 2i^*+1)\}/(l^*, i^*)$$

8:     **if** $t = n$ **then**
        **Return** $x_{l_n, i_n}$ with $(l_n, i_n) \in \arg\max_{\bigcup_{t \geq 1} \mathcal{T}_t} f(x_{l,i})$

---

among all other nodes of the tree search. Although it involves to check the existence of a Lipschitz constant, such computation can easily be executed from a simple geometric perspective. Indeed, a node is potentially optimal if it belongs to the set of Pareto points of the graph representing the levels $l$ of the nodes and their values $f(x_{l,i})$. For more details on this topic, we refer to Appendix E where an illustration and Algorithm 3 to compute the potentially optimal nodes are presented.

## 5.1 The optimistic combinatorial tree search (OCTS) algorithm

The OCTS algorithm (Algorithm 2) implements the optimistic principle [37] over combinatorial trees and aims at optimizing any black-box objective function $f$ defined on the combinatorial space $\mathcal{X} = \{0,1\}^d$. It only takes as input the tree representation $T$ of the search space (Section 3) and the evaluation budget $n$. Similar to OLTS, it starts to evaluate the function $f(x_{0,0})$ at the root node $(0,0)$ of the tree and initiates tree search $\mathcal{T}_1$. At each round, OCTS selects the set of all potentially optimal nodes (line 4) and evaluates the function over their right child nodes (line 6) and updates the tree search (line 7). The main trick here is that since the Lipschitz constant $k \geq 0$ is unknown, we select a larger sets of nodes (i.e., the set of potentially optimal nodes, line 4) that will nonetheless contain the node maximising the upper bound of the OLTS algorithm with known Lipschitz constant. Indeed, denoting by $k_{\min} = \min\{k \geq 0 : f \in \text{Lip}(k)\}$ the smallest (unknown) Lipschitz constant of the objective function, one can easily show that at each round:

$$\arg\max_{(l,i) \in \mathcal{T}_t} \{f(x_{l,i}) + k_{\min}\text{Diam}(\mathcal{X}_{l,i})\} \in \text{OptimalNodes}(\mathcal{T}_t).$$

Thus, with regards to the selection procedure with known Lipschitz constant (line 5 of OLTS), we still select the node that maximises the upper confidence bound with the best possible Lipschitz constant $k_{\min}$ at the extra cost of evaluating the objective function on at most $d-1$ other nodes (one per level) which are also potentially optimal. Although using a similar selection strategy to SOO [37], OCTS manifests three main differences: (1) at each round the selection of potentially optimal nodes is static and does not add the ongoing evaluations of line 6, (2) we do not have a parameter $h(t)$ that controls the maximum depth of the tree, and (3) when a node is selected, we simply switch one component by evaluating the function on its right child and do make several evaluations that depends on the dimensionality of the problem.

## 5.2 Analysis of the OCTS algorithm

As noted above, one of the key features of the algorithm is that at each round, we have the guarantee that it selects the node maximising the upper bound with the best possible Lipschitz constant $k_{\min}$. Therefore, it is possible to recover the following theoretical result:

**Proposition 5.2. (Convergence of OCTS).** *Let $f : \mathcal{X} \to \mathbb{R}$ be any combinatorial function and let $I_l^* := \{node(l,i) \in T : f(x_{l,i}) + k_{\min}Diam(\mathcal{X}_{l,i}) \geq f(x^*)\}$ with $k_{\min} = \min\{k \geq 0 : f \in Lip(k)\}$. Then, if $x_1, \ldots, x_n$ denotes the series of evaluation points generated by OCTS over $f$, we have that*

$$\max_{x \in \mathcal{X}} f(x) - \max_{i=1\ldots n} f(x_i) \leq k_{\min}(d - l(n) - 1)$$

*where $l(n) = \max\{0 \leq L \leq d-1 : (n/d) \geq \sum_{l=0}^{L} |I_l^*|\}$.*

By comparing this results to Proposition 4.3 of Section 4 with known Lipschitz constant, it is interesting to note that here we now use the smallest Lipschitz constant $k_{\min}$ which ensures that $|I_t^*|$ has minimal size while having a division by $d$ in the definition of $l(n)$ which slows down the rate. This trade-off is further detailed in the next result:

**Theorem 5.3. (Fast convergence rates).** *Let $f : \mathcal{X} \to \mathbb{R}$ be any combinatorial function with a unique maximiser and let $c = k_{\min}/k^* < d - 1$ be its conditioning number. Then, if $x_1, \ldots, x_n$ denotes the series of evaluation points generated by OLTC after $n$ iteration, there exists some $n_c \leq d2^{\lceil \frac{2c}{1+2c} d \rceil}$ and $C \leq 2^{\left(\frac{3c}{2c+1}\right)\frac{d}{2}}$ such that:*

$$\max_{x \in \mathcal{X}} f(x) - \max_{t=1...n} f(x_{l_i, i_t}) \leq k_{\min} \times \begin{cases} d - \left\lfloor \frac{\ln(n/d)}{\ln(2)} \right\rfloor, & \text{if } n \leq n_c \\ d - \left\lfloor \frac{\ln(n_c/d)}{\ln(2)} + \frac{n - n_c}{dC} \right\rfloor, & \text{if } n > n_c. \end{cases} \quad (3)$$

Surprisingly, although the Lipschitz constant is not known, the same conclusions still hold to the case when $k$ is known. The main difference is the apparition of a term $d$ which slows down the linear convergence rate at the price of allowing to target objective functions with unknown smoothness.

# 6 Numerical experiments

In this section, we conduct two sets of experiments to 1) show how the theoretical insights provided in the paper are aligned with practice and 2) illustrate the potential of tree search strategies with regard to existing state-of-the-art solvers.

## 6.1 Convergence rates and scaling

First, we compare the empirical performance of OCTS to the baseline Random Search (RS) for which convergence rates are known on three synthetic functions: OneMax $f(x) = \sum_{i=1}^d \mathbb{I}\{x_i = 1\}$, Harmonic $f(x) = \sum_{i=1}^d ix_i$ and LeadingOnes $f(x) = \sum_{i=1}^d \prod_{j=1}^i x_j$. Note that using synthetic functions allows to exactly know the approximation error and rates. Figure 3 displays the approximation error in terms of the number of evaluations averaged over ten runs as well as the number of evaluations required to identify the global optimum. As it can be seen in the first row, the convergence rate of RS is indeed of order $O(-\ln(n))$ as suggested by Proposition B.3 in the Appendix while OCTS achieves an exponentially faster rate of order $O(-n)$ after a certain threshold as suggested by Theorem 5.3. Moreover, as displayed in the second row, it is important to note that this fast convergence allows us to identify the global optimum in at most $O(d^2 \ln(d))$ evaluations instead of $O(2^d)$ evaluations using RS. This, thus, validates the insights provided by the analysis and shows that it is possible to identify the global optimum on problems with large dimension using tree searches.

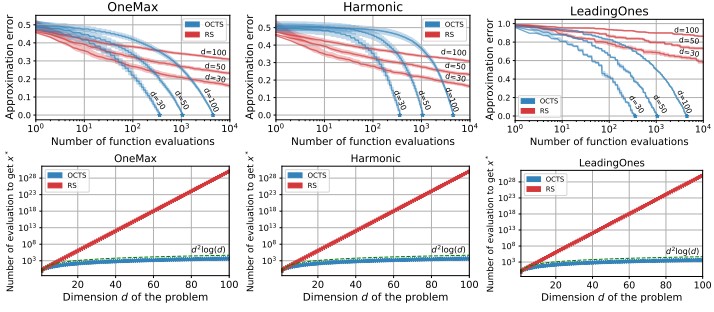

Figure 3: First row: approximation error $\max_{x \in \mathcal{X}} f(x) - \max_{i=1...n} f(x_i)$ in terms of number of function evaluations $n$ in log scale for dimensions 30, 50 and 100 ($\pm 0.1$ standard deviation in transparent). Second row: expected number of function evaluations required to identify the global optimum $x^*$ in log scale in terms of dimension $d$. The curve $d^2 \log(d)$ is plotted in dashed lines for comparison.

## 6.2 Comparison on real-world problems

Here, our goal is to compare the algorithms on several real-world problems with moderate or cheap-to-evaluate objective functions. We compare the performance of OCTS with six methods commonly used to solve combinatorial black-box problems: Simulated annealing [45] (SA), Random Search [9] (RS), Randomized Local Search [38] (RLS), Genetic Algorithm [43] (GA), Evolutionary Algorithm

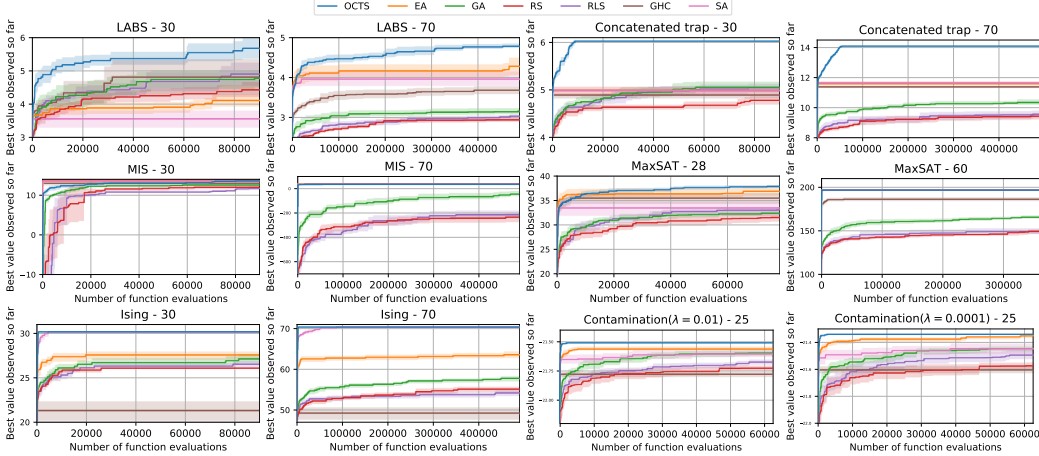

Figure 4: Best valued observed so far in terms of number of function evaluations over ten runs on each problem with $\pm 0.1$ standard deviation in transparent. Title of the plots: problem name - dimension.

[43] (EA) and Greedy Hill-Climbing [27] (GHC). We point out that Bayesian optimization algorithms are omitted from the benchmark due to their heavy computational cost to sample the next evaluation points, as detailed in Appendix F where the computational cost of each methods are provided. Moreover, all the details regarding baselines, test problems, code and additional plots can be found in Appendix F. The algorithms are compared on six problems described below. For each problem, we record the best value $\max_{i=1...t} f(x_i)$ observed at each iteration $1 \leq t \leq n$ over ten runs with various dimensionalities $d$ when possible and an evaluation budget set to $n = 100 \times d^2$ following [18]. Figure 4 displays the results for two dimensions on each problem.

**Low autocorrelation binary sequences (LABS) [17].** Obtaining binary sequences possessing a high merit factor constitutes a grand combinatorial challenge with practical applications in radar engineering and measurements [41, 42]. Given a binary sequence of length $d$, the merit factor is proportional to the reciprocal of the sequence's autocorrelation. The problem consists of searching over the sequence space to yield the maximum merit factor. This hard, non-linear problem has been studied over several decades and still carries open questions concerning its mathematical nature [35, 40]. We ran our algorithms to find the maximum merit factor in various dimensions. As it can be seen, OCTS outperforms all the baselines (see, Figure 4, LABS 30-70). More importantly, OCTS is able to identify sequences with merit factors that none of the other methods could identify.

**Concatenated trap [17].** Concatenated trap is an extension of the LABS problem which consists of partitioning a binary sequence of length $d$ into segments of length $k$ and concatenating $m = d/k$ trap functions that takes each segment as input. Following [17], we considered the trap function $f_{trap}(x) = 1$ if $x$ contains $k$ ones and $(k - 1 - d_H(x, \vec{0}_k))/k$ otherwise with $k = 5$. This is a highly non-trivial optimization problem. Again, OCTS consistently outperforms all competitors and is able to find sequences with values that no other algorithm was able to reach.

**Maximum independent set (MIS) [17].** The MIS problem is the task of finding the largest independent set in a graph, where an independent set is a set of vertices such that no two vertices are adjacent. This problem has applications in network analysis [22] and signal transmission [47], is known to be NP-hard [32]. We compared the algorithms on standard problem instances provided in [5] by maximizing the associated objective of [18]. Here, OCTS is able to identify the best function values in par with other algorithms. However, it is interesting to note that most algorithms perform similarly to RS when $d = 30$, indicating that the problem presents few structure that can be exploited by the different techniques.

**MaxSAT [39].** The satisfiability (SAT) problem is an important combinatorial optimization problem, where one decides how to set variables of a Boolean formula to make the formula true. Many other optimization problems can be reformulated as SAT/MaxSAT problems. Although specialized solvers exist [3], we followed [39] and used MaxSAT as a testbed by running the algorithms on three benchmarks from the MaxSAT Competition 2018. On these problems, OCTS successfully obtains the best score and is able to identify an optimum that no other baseline could find (i.e. MaxSat 28).

**Ising model [17].** The Ising spin glass model arose in solid-state physics and statistical mechanics, aiming to describe simple interactions within many-particle systems. The classical Ising model considers a set of $d$ spins directing up or down placed on a regular lattice, where each edge is associated with an interaction strength. The configuration's energy is described by the system's Hamiltonian, as a quadratic function of those spins variables. The optimization problem is the study of the minimal energy configurations. This is a challenging combinatorial problem, which is known to be NP-hard and holds connections with other NP problems [33]. Again, as it can be seen, the OCTS consistently finds the best configurations in par with SA and is the fastest algorithm.

**Contamination [39].** The contamination control in food supply chain is a binary optimization problem with 25 binary variables [28], where one can intervene at each stage of the supply chain to quarantine uncontaminated food with a cost. The goal is to minimize food contamination while minimizing the prevention cost. We followed the setup of [39] with values of $\lambda = 0.01, 0.0001$ and we observe that OCTS is able to identify novel optimum.

These results show that OCTS consistently outperforms existing methods on a wide range of different problems. More importantly, it also shows that using the novel tree search method allows to discover novel optima that could not previously be reached by existing methods (e.g., LABS, Concatenated).

## 7  Conclusion and Extensions

In this work, we introduced novel combinatorial black-box solvers that rely on tree search. Ideas for future extensions include: (1) designing budget dependent node selection strategies, (2) using a different Lipschitz constant on each dimension and (3) extending the approach to mixed continuous-combinatorial, noisy, contraint and parallel settings.

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
