# Optimistic tree search strategies for black-box combinatorial optimization

**Cedric Malherbe[1], Antoine Grosnit[1], Rasul Tutunov[1], Jun Wang[1,2],
Haitham Bou-Ammar[1,2]**
[1]Huawei Noah's Ark Lab, [2]University College London
`firstname.lastname@huawei.com`

## Abstract

This appendix contains additional material for the paper "Optimistic tree search strategies for black-box combinatorial optimization". In Section B, convergence rates for the baselines are provided. In Section D, we present additional results for the optimization of black-box functions with known Lipschitz constant. Section E introduces an algorithm (Algorithm 3) to compute the set of potentially optimal nodes of the OCTS algorithm. Section F contains the details of the numerical experiments. Finally, Section G contains the proofs of the results of the paper.

## A    Potential negative societal impact

In our work, we proposed a novel methodology to optimize binary functions with cheap-to-evaluate cost. These new solvers are mostly agnostic to the specific application, and can be applied in a wide range of optimization problem (ranging from graph analysis, to electronic design). Therefore, the societal and ethical impacts of our contribution are heavily dependent on the nature of the problems solved with the algorithm.

We start by noting that beneficial applications of OCTS are thick on the ground, ranging from the design a more efficient telecommunication applications, to the control of contaminations, or the design of new solvers for SAT problems. For instance, in the emergence of new pandemics – which could be accelerated by the global warming that entails population displacements, and accelerates the melt of the permafrost threatening to unleash ancient viruses – makes the ability to efficiently plan the allocations of vaccines in a short time more desirable. Another by-product of climate change is the need to make every system energetically efficient to massively curb their energy consumption. The adoption of optimization methods to tune the parameters of energy glutton systems such as data-centers cooling systems, has proven successful in real-world applications.

From the latter example we can perceive that, even though the better tuning of engineering systems make them more accurate or efficient to accomplish their tasks, the positive or negative impact of our work fully depends on the nature of the tasks considered. So, providing novel combinatorial solvers could lead to a more accurate tumor detection model, as it could allow the design of a more racially biased recognition system. However we hope that our contribution alone, as it is incremental, will not in itself encourage individuals to design new malicious models.

Nevertheless, we believe that considering the fundamental challenges we face, starting with the global warming calling for massive optimization efforts as we showed, the overall impact of our work will, in the long run, be positive for the global economy and the well-being of the inhabitants of our planet.

36th Conference on Neural Information Processing Systems (NeurIPS 2022).

# B Convergence rates of the baselines

In this section, we provide some generic results on the convergence of black-box combinatorial solvers and derive convergence results for two of the most famous baselines used to solve Problem (1) in practice: Sequential Exhaustive Search and Pure Random Search. First, it has to be noticed that when only exploring a fraction of the input space (i.e. when $n < |\mathcal{X}|$) even with a carefully designed sequential algorithm, we cannot expect to have a control on the quality of the result provided by any algorithm over the set of all combinatorial black-box functions. Indeed, the next result exhibits this phenomenon.

**Proposition B.1. (No free lunch)** *Let $A$ be any deterministic sequential algorithm defined over $\mathcal{X} = \{0,1\}^d$ and let $C > 0$ be any arbitrarily large constant. Then, for any budget $n < |\mathcal{X}|$ of evaluations, there always exists a function $f_{A,C} : \mathcal{X} \to \mathbb{R}$ such that if one runs the sequential algorithm $A$ over $f$ with an evaluation budget of $n$, we have:*

$$\max_{x \in \mathcal{X}} f_{A,C}(x) - \max_{i=1...n} f_{A,C}(x_i) \geq C$$

*where $x_1, \ldots, x_n$ denotes the set of evaluation points generated by the algorithm $A$ over $f_{A,C}$ after $n$ iterations.*

Hence, any deterministic algorithm might suffer an arbitrarily large error on at least a combinatorial black-box function. To overcome this issue and investigate the finite-time behavior of black-box combinatorial solvers, a possible strategy consists on focusing on a smaller set of combinatorial functions through the lens of their Lipschitz constant, i.e., the value $k \geq 0$ such that $|f(x) - f(x')| \leq k d_H(x, x')$ for all $(x, x') \in \mathcal{X}^2$. This coefficient $k \geq 0$ provides an information on the smoothness of the function in the sense that it captures how fast the objective function can vary. More importantly, due to the discrete nature of combinatorial spaces, it is important to note that every combinatorial function has a finite Lipschitz constant given by $k_{\min} = \max_{x \neq x' \in \mathcal{X}^2} |f(x) - f(x')|/d_H(x, x')$ which also appears in the theoretical analysis of OCTS (Theorem 5.3). Thus, when considering the set of $k$-Lipschitz functions with a fixed $k \geq 0$, one can derive a finite-time analysis of the baseline strategies.

**Proposition B.2. (Convergence of sequential exhaustive search).** *A sequential exhaustive search simply consists in sequentially evaluating the function over the whole space $\mathcal{X}$ in any specific order. As an example, we consider the exhaustive search which consists in exploring the tree representation of the space provided in Section 3 in a breadth-first search manner. For this strategy, one can show that for any function $f \in Lip(k)$ and any budget $1 \leq n \leq 2^d$, we have:*

$$\max_{x \in \mathcal{X}} f(x) - \max_{i=1...n} f(x_i) \leq k \left( d - \left\lfloor \frac{\ln(n)}{\ln(2)} \right\rfloor \right)$$

*where $x_1, \ldots, x_n$ denotes the series of points generated be the exhaustive search.*

**Proposition B.3. (Convergence of random search).** *The second baseline, Random Search, consists in evaluating the objective function on a series of $n$ candidates $x_1, \ldots, x_n \sim \mathcal{U}(\mathcal{X})$ uniformly distributed over the input space. For this strategy, one can show that for any function $f \in Lip(k)$ and any budget $2^{d/2} \leq n \leq 2^d$, we have with probability at least $1 - 1/e$:*

$$\max_{x \in \mathcal{X}} f(x) - \max_{i=1...n} f(x_i) \leq k \left( d - \left\lfloor \frac{\ln(n)}{\ln(2)} \right\rfloor \right).$$

To the best of our knowledge, it the first time these sample complexity results are reported for combinatorial black-box problems. It is important note that the convergence rates of both these strategies are of order $k(d - \ln(n)/\ln(2))$, as they play the role of baselines in our analysis. Finally, we cast a simple result related to the number of iterations required to identify a global optimum using random search.

**Proposition B.4. (Consistency of random search).** *Consider the Random Search (RS) and let $x_1, x_2, x_3, \ldots$ denotes a series of points independently and uniformly distributed over the input space $\mathcal{X} = \{0,1\}^d$. Then, for any function $f : \mathcal{X} \to \mathbb{R}$, if $N^* = \min\{t \geq 1 \mid f(x_t) = \max_{x \in \mathcal{X}} f(x)\}$ denotes the number of function evaluations required to identify a global optimum $x^* \in \mathcal{X}^* := \arg\max_{x \in \mathcal{X}} f(x)$, we have that*

$$\mathbb{E}[N^*] = \frac{2^d}{|\mathcal{X}^*|}$$

*where $|\mathcal{X}^*|$ denotes the number of global optima of the function $f$.*

This result will be used in the numerical experiments section to compare our algorithms.

## C    Comparison of combinatorial vs. continuous trees

Here, we list a difference between the trees introduced in the paper 3 and the trees introduced to optimize continuous structures [37]. Keeping in mind that the nature of those trees are really different, we list here the following differences. We call combinatorial trees the trees we introduced in Section 3 and we call continuous trees, the trees introduced in [37].

- Width. In combinatorial trees, only one coordinate is switched at each split. Doing the same in continuous trees would result in losing the decreasing diameter property (Assumption 3.3). As a consequence, combinatorial trees only have 2 children per node (independently of the dimension) while in continuous trees we have $2^d$ children per node. Thus, continuous trees are much wider/flat trees that exponentially explode with the dimension. Moreover, since combinatorial trees impose that the left child has the same value as its parent node, an import consequence is that one can easily navigate through the tree linearly with $d$ evaluations while it would require $d2^d$ evaluations in continous trees, which explodes with the dimension $d$. Note that this trick imposes that the size of combinatorial trees is $2^{d+1}$ instead of $2^d$.

- The depth of combinatorial trees is $d + 1$ while the depth of ContTree is infinite. In practice, continuous trees are controlled by a parameter $h$ which limits its depth at a given time and impacts the performance of the search. In [36] they obtain two very distinct regimes of convergence that depend on the parameter $h$ (exponential and polynomial), while we only obtain a single (fast) linear regime of convergence.

- the nodes of combinatorial trees can be represented as $x_{l,i} = bin_l(i) + \vec{0}_{d-l}$. It allows to simply store the index $(l, i)$ of the tree search instead of the full vectors $x_{l,i}$ of dimension $d$ for a better scaling w.r.t. the dimension

- From a theoretical perspective, most of the analysis boils down to bounding the volume of the sphere $B(x_{l,i}, R)$ for some $R > 0$ where $x_{l,i}$ is any point in the tree. In continuous space, it is easy to integrate and proportional to $R^d$, while in combinatorial spaces, the results are discrete (hence $l(n)$) and (combinatorially) explode with $R$. To overcome this phenomenon, we introduce specific combinatorial techniques (see the proofs of the lemmas).

## D    Additional results for the OLTS algorithm (Section 4)

Here, we provide an additional analysis of OLTS. First, we start to explain the mechanisms behind the optimistic node selection policy (line 5) based on the general analysis of optimistic searches [37]. Let $x^* \in \arg\max_{x \in \mathcal{X}} f(x)$ be any maximizer of the objective function. By construction of the search tree, we know that at any time $t \geq 1$, if a global maximizer $x^*$ has not been identified yet, there necessarily exists a node$(l_t^\star, i_t^\star)$ in the current tree search $\mathcal{T}_{t-1}$ that contains the global optimum as a child (i.e., $x^\star \in \mathcal{X}_{l_t^\star, i_t^\star}$). Moreover, by definition of the selection policy and since $f \in \text{Lip}(k)$, we know that:

$$\max_{(l,i) \in \mathcal{T}_{t-1}} f(x_{l,i}) + k\text{Diam}(\mathcal{X}_{l,i}) \geq f(x_{l_t^\star, i_t^\star}) + k\text{Diam}(\mathcal{X}_{l_t^\star, i_t^\star})$$

$$\geq f(x_{l_t^\star, i_t^\star}) + kd_H(x^*, x_{l_t^\star, i_t^\star})$$

$$\geq f(x^*).$$

Thus, the algorithm will never select nodes from the tree such that $f(x_{l,i}) + k\text{Diam}(\mathcal{X}_{l,i}) < f(x^*)$ until it identifies a global optimum. As a consequence, the algorithm will only select nodes in the set $I = \bigcup_{l=0}^{d-1} I_l$ where

$$I_l := \{\text{nodes}(l, i) \in T \text{ s.t. } f(x_{l,i}) + k\text{Diam}(\mathcal{X}_{l,i}) \geq f(x^*)\} \tag{4}$$

until it finds the global optimum. Based on this observation, we may now formulate a result which connects the size of the sets $|I_l|$ to the convergence of the algorithm. Note that it completes Proposition D.1 provided in the paper.

**Proposition D.1. (Convergence of the OLTS Algorithm)** *Consider any $f \in Lip(k)$ and let $x_1, \ldots, x_n$[3] denote the set of evaluations points generated by OLTS after $n$ iterations over the function $f$ tuned with Lipschitz constant $k$ and the tree representation of Section 3. Then,the minimum number of iterations $n^*$ required to identify the global optimum is upper bound as follows:*

$$n^* := \min\left\{n \leq 2^d : f(x_n) = \max_{x \in \mathcal{X}} f(x)\right\} \leq 1 + \sum_{l=0}^{d-1} |I_l|$$

*Moreover, for any $n < n^*$, we have the following finite-time bound:*

$$\max_{x \in \mathcal{X}} f(x) - \max_{i=1\ldots n} f(x_i) \leq k \cdot (d - l(n))$$

*where $l(n) := \min\{0 \leq L \leq d - 1 : \sum_{l=0}^{L} |I_l| \geq n - 1\}$.*

As explained in Section 4, we can derive a finer analysis of the convergence of the algorithm by using the conditioning number defined as follows.

**Definition D.2. (Conditioning Number).** Let $f : \mathcal{X} \to \mathbb{R}$ be any $k$-Lipschitz function. Then, if $f$ admits a unique global maximum $x^* \in \mathcal{X}$, we denote by $k^* \geq 0$ the largest value such that, for all $x \in \mathcal{X}$, we have:

$$f(x) \leq f(x^*) - k^* d_H(x, x^*)$$

and we denote by $c = k/k^* \geq 1$ its conditioning number.

It is also interesting to note that this definition can be extended to functions with multiple local optimum by taking the smallest $k^*$ among the local optima resulting in additional factors in the next lemma. Based on this number, we can now make the link between the size of $I_l$ and the conditioning number.

**Lemma D.3.** *Let $f \in Lip(k)$ be any objective function with a unique maximizer and denote by $c = k/k^*$ its conditioning number. Then, if $I_l$ denotes the set defined in (4) at level $l$, we have that:*

$$|I_l| \leq \sum_{i=0}^{\lfloor c(d-l) \rfloor \wedge l} \binom{l}{i}.$$

It is interesting to compare this result to the exhaustive search (Appendix B) where at each level $l \geq 0$, the function is evaluated on $2^l = \sum_{i=0}^{l} \binom{l}{i}$ points. In this case, we will have a gain as soon as there exists $l \leq d - 1$ such that $\lfloor c(d - l) \rfloor < l$. This translates into the condition $c = k/k^* < d - 1$ and we deduce that if the Lipschitz constant $k$ is set larger than $k^*(d - 1)$ we will not get any gain over an exhaustive search. This might not be surprsing as when $k$ is too large the upper bound in line 5 provides a loose information. However, it is possible to obtain a finite-time bound when $k < k^*(d - 1)$ by upper bounding the partial sums of binomial coefficients, which unfortunately does not admit a closed form.

**Lemma D.4.** *Let $d \geq 2$ and set any $c \geq 1$. Then, for any $2cd/(1 + 2c) \leq l \leq d$, we have that*

$$\sum_{i=0}^{\lfloor c(d-l) \rfloor} \binom{l}{i} \leq 2^{\left(\frac{3c}{2c+1}\right)\frac{d}{2}}.$$

Again, this Lemma allows to provide a bound on the size of the set $|I_l|$ that does not depends on the level $l$ and does not explode at the rate $2^l$ provided by the exhaustive search. Using this result, we can prove Theorem 4.3 (see Section G).

**Example.** As an example, when $f(x) = -d_H(x, \vec{1})$ with the tree of Section 3 and OLTS with $k = 1.5$ and $d = 4$, we have $|I_0| = 1, |I_1| = 2, |I_2| = 3, |I_3| = |I_4| = 1$. Thus, in this case, we have $l(1) = 0, l(4) = 1, l(7) = 2$ and $l(8) = 3$ and we know that after $8$ iterations, we will be at least at the level 3.

---

[3] $x_1 = x_{0,0}$ and $x_t = x_{l_t+1, 2i_t+1}$ for $t \geq 2$

# E    Computation of the potentially optimal nodes for OCTS (Section 5)

In this section, we provide an algorithm to compute the set of potentially optimal nodes. Recall first that the set of potentially optimal nodes is defined as follows.

**Definition E.1. (Potentially optimal nodes)** Let $\mathcal{T} = \{(l_1, i_1), \ldots, (l_t, i_t)\}$ be any set of index of the tree where we have an evaluation of the objective function $f(x_{l_i, i_i})$. Then, a node $(l, i)$ from the tree search $\mathcal{T}$ is said to be potentially optimal if there exists some Lipschitz constant $k > 0$ such that for all $(l', i') \in \mathcal{T}$:

$$f(x_{l,i}) + k\text{Diam}(\mathcal{X}_{l,i}) \geq f(x_{l',i'}) + k\text{Diam}(\mathcal{X}_{l',i'}).$$

In order to further characterize this set, observe first that for any level $l \in \{1, \ldots, d-1\}$, if there exists $(i, i') \in \{0, \ldots, 2^l\}^2$ such that $f(x_{l,i}) > f(x_{l,i'})$ then for all $k \geq 0$ we have $f(x_{l,i}) + k\text{Diam}(\mathcal{X}_{l,i}) > f(x_{l,i'}) + k\text{Diam}(\mathcal{X}_{l,i'})$. Thus, in this case, we deduce that $f(x_{l,i'})$ does not belong to the set of potentially optimal nodes. Moreover, if there exists $(l, i) \in \mathcal{T}_t$ and $(l', i') \in \mathcal{T}_t$ with $l' < l$ and $f(x_{l',i'}) > f(x_{l,i})$, then $f(x_{l,i}) + k\text{Diam}(\mathcal{X}_{l,i}) < f(x_{l,i'}) + k\text{Diam}(\mathcal{X}_{l,i'})$ for all $k \geq 0$ and we deduce that $(l, i)$ is not potentially optimal. As a consequence, we deduce that the set of potentially optimal nodes necessarily belongs to the Pareto front of the graph displaying the level of the node $l$ and the function value $f(x_{l,i})$. Let us now only consider the set of points that belong the Pareto front. Then, for any level $l' < l$, we have $f(x_{l,i}) + k\text{Diam}(\mathcal{X}_{l,i}) \geq f(x_{l',i'}) + k\text{Diam}(\mathcal{X}_{l',i'})$ whenever $k \leq (f(x_{l,i}) - f(x_{l',i'}))/(l - l')$. Moreover, for any level $l' > l$, we have $f(x_{l,i}) + k\text{Diam}(\mathcal{X}_{l,i}) \geq f(x_{l',i'}) + k\text{Diam}(\mathcal{X}_{l',i'})$ whenever $k \geq (f(x_{l',i'}) - f(x_{l,i}))/(l' - l)$. Thus, we deduce that node $(l, i)$ on the Pareto front is potentially optimal if and only if $\max_{l'>l}(f(x_{l',i'}) - f(x_{l,i}))/(l' - l) \leq \min_{l'<l}(f(x_{l,i}) - f(x_{l',i'}))/(l-l')$. We can now derive a simple algorithm (Algorithm 3) to compute the set of potentially optimal nodes. Moreover, Figure 5 gives an example of the computation of this set.

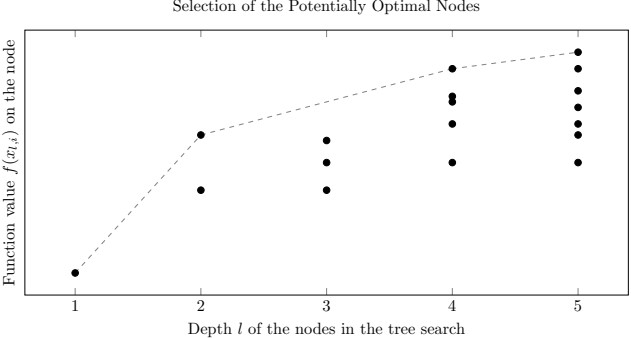

Selection of the Potentially Optimal Nodes

Figure 5:  Example of a the computation of the potentially optimal nodes on a toy example with $d = 6$. The graph displays the values of the function and the depth of the nodes $(l, i)$ of the search list $\mathcal{T}_t$. The four nodes selected are located on the pareto front.

# F    Numerical experiments

In this section, we provide the details on regarding the test functions, algorithms and results of the empirical results of the paper. Note that the code of the numerical experiments can be found at: https://anonymous.4open.science/r/combinatorial_opt_ICML-71ED. We point out that, in our experience, the link is sometimes unexpectedly off (it may be due to the fact the anonymization service is free). However, in the worst case the code will nonetheless be available upon acceptance.

## F.1    Baselines

Here, we describe the algorithms used in the experimental section.

**Random Search (RS), Algorithm 4.** This algorithm simply consists in evaluating the value of the function on a series of evaluation points generated randomly over the input space $\{0, 1\}^d$.

**Evolutionary algorithm (EA), Algorithm 5.** In evolutionary computation, an initial set of candidate solutions is generated and iteratively updated. We used the evolutionary algorithm provided in

---

**Algorithm 3** Optimal Nodes

---

**Require:** Tree search nodes $\mathcal{T} = \{(i_1, l_1), \ldots, (i_{|\mathcal{T}|}, l_{|\mathcal{T}|})\}$
1: $\mathcal{T}^* \leftarrow \emptyset$
2: $f_{\max} = -\infty$
3: **for** $l = 0 \ldots d - 1$ **do**
4:     Get a node at level $l$ with the largest value among all the nodes of level $l$ if there is one (break ties arbitrarily):
$$(l, i^*) \in \underset{(l', i) \in \mathcal{T}: l' = l}{\arg \max} f(x_{l', i})$$
5:     **if** $f(x_{l, i^*}) \geq f_{\max}$ **then**
6:         Add the node to the optimal node list and update the $f_{\max}$ value
7:         $\mathcal{T}^* \leftarrow \mathcal{T}^* \cup \{(l, i^*)\}$
8:         $f_{\max} \leftarrow f(x_{l, i^*})$
9: **for** $(l, i) \in \mathcal{T}^*$ **do**
10:    **if** $\max_{(l', i') \in \mathcal{T}^*: l' > l}(f(x_{l', i'}) - f(x_{l, i}))/(l' - l) > \min_{l' < l}(f(x_{l, i}) - f(x_{l', i}))/(l - l')$ **then**
        $\mathcal{T}^* \leftarrow \mathcal{T}^*/\{(l, i)\}$
11: **Return:** $\mathcal{T}^*$

---

---

**Algorithm 4** Random Search (RS)

---

**Require:** evaluation budget $n \geq 1$
1: **for** $t = 1, \ldots n$ **do**
2:     Let $x_t \sim \mathcal{U}(\{0, 1\}^d)$
3:     Evaluate $f(x_t)$
4: **Return** $x_{t^*}$ where $t^* \in \arg\max_{t=1\ldots n} f(x_t)$

---

[18]. At each time step, $\lambda = 10$ mutations are performed from a current incumbent $x^*$ with a static mutation rate $p = 1/d$ and using a binomial distribution. The mutation phase is defined by sampling around the current incumbent using a binomial variable. At the end of the mutations (lines 5-8), we switch to a novel if the offspring has better function values than the current incumbent (line 11). In all the experiments, $x_{init} \sim \mathcal{U}(\{0, 1\}^d)$ was uniformly sampled over the input domain.

---

**Algorithm 5** Evolutionary Algorithm (EA)

---

**Require:** evaluation budget $n \geq 1$, $x_{init} \in \{0, 1\}^d$
1: $x^* \leftarrow x_{init}$
2: Evaluate $f(x^*)$, $t \leftarrow 1$
3: **while** $t \leq n$ **do**
4:     **for** $i = 1, \ldots \lambda$ **do**
5:         Sample $m = \text{Bin}(d, 1/d)$
6:         Let $x_{t+1} = x^* + m$
7:         Evaluate $f(x_{t+1})$, $t \leftarrow t + 1$
8:         **if** $t = n$ **then**
            **Return** $x_{t^*}$ with $t^* \in \arg\max_{t=1\ldots n} f(x_i)$
9:     $x^*_{new} \leftarrow \arg\max_{x \in x_{t-\lambda}, \ldots, x_t} f(x)$
10:    **if** $f(x^*_{new}) \geq f(x^*)$ **then**
11:        $x^* \leftarrow x^*_{new}$

---

**Genetic Algorithm (GA), Algorithm 6.** Genetic Algorithms use mechanisms inspired by biological evolution, such as reproduction, mutation, recombination, and selection to optimize black-box functions [44] The binary crossover-based evolutionary of [18] was used in the testbed. The selection, crossover and mutation operations are defined as follows. First, we sample an initial population of $\lambda = 30$ points sampled at random in the input space. Then, we perform a series of selection, crossover, mutation until the evaluation budget is reached. For the selection phase (line 7), we select $\lambda$ elements $x_{t+1}, \ldots, x_{t+\lambda}$ from the existing points $x_1, \ldots, x_t$ by sampling the points without replacement from the softmax of the function values, (i.e. the probability of selecting individual $i$ is equal to $e^{f(x_i)}/\sum_{i=1\ldots t} e^{f(x_i)}$). For the crossover, we consider each pair $(x_{t+i}, x_{t+i+\lambda/2})$ with

$i = 1 \ldots \lambda/2 - 1$ and perform a crossover with probability $p = 0.37$. For a crossover, a random index $j \sim \mathcal{U}([1, \ldots, d])$ is sampled and we perform the operations $x_{t+i} \leftarrow x_{t+i}[1, \ldots, j] + x_{t+\lambda/2+i}[j+1, \ldots, n]$ and $x_{t+\lambda/2+i} \leftarrow x_{t+\lambda/2+i}[1, \ldots, j] + x_{t+i}[j+1, \ldots, n]$. Finally, for the mutation phase, we select each candidate $x_t + i$ with $i = 1 \ldots \lambda$ and flip each of its bit with probability $1/2d$.

---

**Algorithm 6** Genetic Algorithm (GA)

---

**Require:** evaluation budget $n \geq 1$
1: $\lambda = 30$
2: **for** $t = 1 \ldots, \lambda$ **do**
3: $\quad x_t \leftarrow \mathrm{Bin}(d, 1/2)$
4: $\quad$ Evaluate $f(x_t)$
5:
6: **while** $t \leq n$ **do**
7: $\quad x_{t+1}, \ldots, x_{t+\lambda} \leftarrow \mathrm{SOFTMAX}(f(x_1), \ldots, f(x_t), \lambda)$
8: $\quad x_{t+1}, \ldots, x_{t+\lambda} \leftarrow \mathrm{Crossover}(x_{t+1}, \ldots, x_{t+\lambda})$
9: $\quad x_{t+1}, \ldots, x_{t+\lambda} \leftarrow \mathrm{Mutation}(x_{t+1}, \ldots, x_{t+\lambda})$
10: $\quad$ **for** $i = 1 \ldots \lambda$ **do**
11: $\quad\quad$ Evaluate $f(x_{t+1})$, $t \leftarrow t + 1$
12: $\quad\quad$ **if** $t = n$ **then**
13: $\quad\quad\quad$ **Return** $x_{t^*}$ with $t^* \in \arg\max_{t=1\ldots n} f(x_i)$

---

**Simulated annealing (SA), Algorithm 7.** The name of the algorithm comes from annealing in metallurgy, a technique involving heating and controlled cooling of a material to alter its physical properties. At each step, the simulated annealing heuristic considers some neighboring state $x_{new}$ of the current state $x^*$, and probabilistically decides between moving the system to state $x_{new}$ or staying in-state $x^*$. These probabilities ultimately lead the system to move to states of lower energy. Typically this step is repeated until the computation budget has been exhausted. The probability of switching to a new state (also called the annealing schedule) was taken from [39]. The algorithm is presented in Algorithm 7. In all the experiments, $x_{init} \sim \mathcal{U}(\{0,1\}^d)$ was uniformly sampled over the input domain.

---

**Algorithm 7** Simulated Annealing (SA)

---

**Require:** evaluation budget $n \geq 1$, $x_{init} \in \{0,1\}^d$
1: $x_1 \leftarrow x_{init}$
2: Evaluate $f(x_1)$
3: $x^* \leftarrow x_1$
4: $T = 10$
5: **for** $t = 2, \ldots, n$ **do**
6: $\quad i \sim \mathcal{U}(1, \ldots, d)$
7: $\quad x_t = x^*$
8: $\quad x_t[i] = 1 - x_t[i]$
9: $\quad$ Evaluate $f(x_t)$
10: $\quad$ **if** $e^{(f(x_t) - f(x^*))/T} \geq \mathcal{U}([0,1])$ **then**
11: $\quad\quad x^* \leftarrow x_t$
12: $\quad T \leftarrow T \times e^{-1/d}$
13: **Return:** $x_{t^*}$ where $t^* \in \arg\max_{t=1\ldots n} f(x_t)$

---

**Greedy Hill-Climber (GHC), Algorithm 8.** Hill climbing is an optimization technique which belongs to the family of local searches. It is an iterative algorithm that starts with an arbitrary solution to a problem, then attempts to find a better solution by making an incremental change to the solution. If the change produces a better solution, another incremental change is made to the new solution, and so on until no further improvements can be found. The Greedy Hill Climber is a hill climber that goes from the current solution from left to right, flipping exactly one bit per each iteration, and accepting to keep the bit switch if it is at least as good as the current vector. In all the experiments, $x_{init} \sim \mathcal{U}(\{0,1\}^d)$ was uniformly sampled over the input domain.

**Randomized Local Search (RLS), Algorithm 9.** Randomized Local Search (RLS), is an ellitist strategy flipping one uniformly chosen bit in each iteration. Here, the main difference between RLS

---
**Algorithm 8** Greedy Hill-Climber (GHC)
---
**Require:** evaluation budget $n \geq 1$, $x_{init} \in \{0, 1\}^d$
1: $x_1 \leftarrow x_{init}$
2: Evaluate $f(x_1)$
3: $x^* \leftarrow x_1$
4: **for** $t = 2, \ldots, n$ **do**
5:     $i = 1 + t \bmod d$
6:     $x_t = x^*$
7:     $x_t[i] = 1 - x_t[i]$
8:     Evaluate $f(x_t)$
9:     **if** $f(x_t) \geq f(x^*)$ **then**
10:       $x^* \leftarrow x_t$
11: **Return:** $x_{t^*}$ where $t^* \in \arg\max_{t=1\ldots n} f(x_t)$
---

and GHC is that here the flipped bit is chosen at random while it is deterministic in GHC. In all the experiments, $x_{init} \sim \mathcal{U}(\{0, 1\}^d)$ was uniformly sampled over the input domain.

---
**Algorithm 9** Randomized Local Search (RLS)
---
**Require:** evaluation budget $n \geq 1$, $x_{init} \in \{0, 1\}^d$
1: $x_1 \leftarrow x_{init}$
2: Evaluate $f(x_1)$
3: $x^* \leftarrow x_1$
4: **for** $t = 2, \ldots, n$ **do**
5:     $i \leftarrow \mathcal{U}([1, \ldots, d])$
6:     $x_t = x^*$
7:     $x_t[i] = 1 - x_t[i]$
8:     Evaluate $f(x_t)$
9:     **if** $f(x_t) \geq f(x^*)$ **then**
10:       $x^* \leftarrow x_t$
11: **Return:** $x_{t^*}$ where $t^* \in \arg\max_{t=1\ldots n} f(x_t)$
---

**Optimistic Combinatorial Tree Search.** It is the algorithm described in Algorithm 2. For this algorithm, we used the tree of Section 3 with a root node randomly sampled over the input space.

Note that Bayesian methods are not included in the benchmark since they simply cannot be run with the allocated budget of the experiments. In particular, in [39], they report that their Bayesian implementation can perform 270 evaluations in 10 hours, which makes them unrealistic to scale on the problems we consider (on at least 40000 function evaluations).

### F.2 Computational cost of the algorithms at each iteration

In addition to the description of the algorithms provided above, we provide here the computational time required to sample the next evaluation point $x_{t+1}$ at each iteration $t \geq 1$. The table below reports the computational cost of each method.

| | OCTS | GA | EA | RS | RLS | GHC | SA | Bayesian |
|---|---|---|---|---|---|---|---|---|
| Complexity to sample $x_{t+1}$ | $O(d)$ | $O(t)$ | $O(1)$ | $O(1)$ | $O(1)$ | $O(1)$ | $O(1)$ | $O(2^d)$ |
| Memory to compute $x_{t+1}$ | $t+1$ | $t+1$ | $\lambda(30)$ | 1 | 2 | 2 | 2 | $t+1$ |
| Time (s) to compute $x_{t+1}$ | 0.001 | 0.004 | 0.001 | 0.001 | 0.001 | 0.0008 | 0.0007 | 62.00 |

Table 1: Computational complexity of different algorithms

Note that the time to compute $x_{t+1}$ is measured on the contamination problem (d=25) on a i7 CPU @ 1.80GHz 1.99 GHz with 16GB of RAM after $t = 100$ iterations over the Contamination problem. As

it can be seen, OCTS is in the same order of magnitude as other methods and significantly faster than Bayesian methods. For the Bayesian optimization method, we took COMBO [39] as well as their official implementation. Moreover, we point out that the complexity of maximising the acquisition function in Bayesian optimization is equivalent to solving in practice a combinatorial black-box optimization. Hence, the complexity of $O(2^d)$ reported in the table. However, we point out that in most implementations, approximate methods such as simulated annealing are used.

### F.3 Test problems

In this section, we provide a full description of the test problems of the benchmark.

**OneMax (from [18]).** The OneMax is probably the best-studied benchmark problem in the context of discrete optimization. It asks to optimize the function

$$f(x) = \sum_{i=1}^{d} x_i.$$

The problem has a very smooth and non-deceptive objective landscape. It is separable and admits a unique maximum located in $x^* = \vec{1}_d$. Due to the so-called coupon collector effect, it is relatively easy to make progress when the function values are small, and the probability to obtain an improving move decreases considerably with increasing function value. Note that this function is 1-Lipschitz with $k_{\min} = 1$ and has a conditioning number equal to $k$. This function was coded in our benchmark.

**Harmonic (from [18]).** Two extreme linear functions are OneMax (presented above) with its constant weights and binary value $\sum_{i=1}^{d} 2^i x_i$ with its exponentially decreasing weights. An intermediate linear function is the Harmonic function:

$$f(x) = \sum_{i=1}^{d} i x_i$$

Again, this problem is separable and admits a unique maximizer $x^* = \vec{1}_d$. The function has $k_{min} = d$ and is $d$-Lipschitz. It has a conditioning number equal to $k/d$. This function was coded in our benchmark.

**LeadingOnes (from [18]).** The LeadingOnes function is certainly the one receiving most attention in the theory of evolutionary community. This problem asks to maximize the function:

$$f(x) = \max\{i \in [0, \dots, n] \mid \forall j \leq i : x_j = 1\} = \sum_{i=1}^{n} \prod_{j=1}^{i} x_j$$

which counts the number of initial ones. This function is non-separable and has a unique maximizer $x^* = \vec{1}_d$. It was coded in our benchmark.

**LABS (from [18]).** Obtaining binary sequences possessing a high merit factor, also known as the LowAutocorrelation Binary Sequence (LABS) problem, constitutes a grand combinatorial challenge with practical applications in radar engineering and measurements. It poses a non-linear objective function over a binary sequence space, with the goal to maximize the reciprocal of the sequence's autocorrelation:

$$f(x) = \frac{d^2}{E(x)} \text{ where } E(x) = \sum_{k=1}^{n-1} \left( \sum_{i=1}^{n-k} s_i \times s_{i+k} \right)^2$$

with $s_i$ set to $s_i = 2x_i - 1$ to cast the problem in $\{-1, 1\}^d$ when $x_i \in \{0, 1\}^d$. This hard, non-linear problem has been studied over several decades, where the only way to obtain exact solutions remains exhaustive search. For this problem, we took the implementation of [18][4].

**Concatenated Trap (from [18]).** Concatenated trap is defined by partitioning a bit-string into segments of length $k$ and concatenating $m = n/k$ trap functions that takes each segment as input.

---

[4]https://iohprofiler.github.io/

The trap function is defined as follows: $f_{trap}(x) = 1$ if $x$ contains $k$ ones and $(k - 1 - d_H(x, \vec{0}_k))/k$ otherwise with $k = 5$. Thus, the optimization objective can be formulated as follows:

$$f(x) = \sum_{i=0}^{m-1} f_{trap}(x_{ik:(i+1)k})$$

where $x_{i:j}$ denotes the elements from $i$ to $j$ of the vector $x$. For this problem, we took the implementation of [18][4].

**Maximum Independent Set (from [18]).** Given a graph $G = (V, E)$, a maximum independent vertex set (which generally is not equivalent to a maximal independent vertex set) is a subset of vertices where no two vertices are are direct neighbors. A maximum independent vertex set (MIS) is defined as an independent vertex set $V'$ having largest possible size. Using the standard binary encoding $V' = \{i = 1 \ldots d \mid x_i = 1\}$, MIS can be formulated as follows:

$$f(x) = \sum_{i=1}^{d} x_i - d \sum_{i,j} x_i x_j e_{i,j}.$$

where $e_{i,j} = 1$ if $(i, j) \in E$ and 0 otherwise. In particular, following [5], a specific, scalable problem instance, defining its Boolean graph was considered and defined as follows:

$$e_{i,j} = 1 \Leftrightarrow j = i + 1 \ \forall i \in \{1, \ldots d\} - \{d/2\}$$
$$\text{or } j = i + d/2 + 1 \ \forall i \in \{1, \ldots, d/2 - 1\}$$
$$\text{or } j = i + d/2 - 1 \ \forall i \in \{2, \ldots, d/2\}.$$

For this problem, we took the implementation from [18][4].

**MaxSAT (from [39]).** Satisfiability problem is the one of the most important and general form of combinatorial optimization problems. SAT solver competition is held in Satisfiability conference every year[5]. We followed [39] and used the three benchmarks of weighted maximum satisfiability problems with no hard clause with the number of variables not exceeding 100. The weights are normalized by mean subtraction and standard deviation division. For the MaxSAT problem, we directly took the implementation and description provided in [39][6].

**Ising Problem (from [18]).** The Ising Spin Glass model arose in solid-state physics and statistical mechanics, aiming to describe simple interactions within many-particle systems. The classical Ising model considers a set of spins placed on a regular lattice, where each edge $< i, j >$ is associated with an interaction strength $J_{i,j}$. In essence, a problem-instance is defined upon setting up the coupling matrix $J_{i,j}$. Each spin directs up or down, associated with a value $\pm 1$, and a set of $d$ spin glasses is said to form a configuration, denoted as $S = (s_1, \ldots, s_d)$. The configuration's energy function is described by the system's Hamiltonian, as a quadratic function of those $d$ spin variables: $-\sum_{i<j} J_{i,j} s_i s_j - \sum_{i=1}^{d} h_i s_i$ where $h_i$ is an external magnetic field. The optimization problem of interest is the study of the minimal energy configurations, which are termed ground states, on a final lattice. The implementation of the Ising model of [18] was taken, assuming zero external magnetic fields, and applying periodic boundary conditions (PBC). To formally define the objective function, we adopt a strict graph perspective, where $G = (V, E)$ is undirected and $|V| = d$. We apply an affine transformation $\{-1, +1\} \rightarrow \{0, 1\}$ where the $d$ spins become binary decision variables. A generalized, compact form for the quadratic objective function is written as follows:

$$f(x) = \sum_{(u,v) \in E} [x_u x_v - (1 - x_u)(1 - x_v)]$$

where the graph $G$ is defined as follows $e_{i,j} = 1$ if and only if $j = i + 1$ for all $i \in \{1, \ldots, d - 1\}$ or $j = d$ and $i = 1$. For this problem, we took the implementation of [18][4].

**Contamination (from [39]).** The contamination control problem ([28]) considers a food supply with $d = 25$ stages that may be contaminated with pathogenic microorganisms. The problem is about minimizing the contamination of food where at each stage a prevention effort can be made

---

[5]http://sat2018.azurewebsites.net/competitions/
[6]https://github.com/QUVA-Lab/COMBO

to decrease a possible contamination. Specifically, we let random variable $Z_i$ denote the fraction of contaminated food at stage $i$ for $1 \leq i \leq d$. At each stage $i$, a prevention effort can be made to decrease the contamination by a random rate $\Gamma_i$ incurring a cost $c_i$. If no prevention effort is taken, the contamination spreads with rate given by random variable $\Delta_i$. This results in the recursive equation $Z_i = \Delta_i(1 - x_i)(1 - Z_{i-1}) + (1 - \Gamma_i x_i)Z_{i-1}$, where $x_i \in \{0, 1\}$ is the decision variable associated with the prevention effort at stage $i$. Thus, the goal is to decide for each stage whether to implement a prevention effort in order to minimize the cost while ensuring the fraction of contaminated food does not exceed an upper limit $U_i$ with probability at least $1 - \varepsilon$. The random variables $\Delta_i, \Gamma_i$ and the initial contamination fraction $Z_i$ follow beta-distributions, whereas $U_i = 0.1$ and $\varepsilon = 0.05$. Here, the problem consists of minimizing the Lagrangian relaxation of the problem:

$$\arg\min_{x \in \{0,1\}^d} \sum_{i=1}^{d} \left[ c_i x_i + \frac{\rho}{T} \sum_{k=1}^{T} \mathbb{I}\{Z_k > U_i\} \right] + \lambda \|x\|_1$$

where each violation is penalized by $\rho = 1$. The regularization term encourages the prevention efforts to occur at a small number of stages. For this problem, we turned the problem into maximization by multiplying the objective function by $(-1)$. We used the implementation of [39][6] where $T$ is set to 100, the dimensionality $d$ to 25 and

## F.4 Experimental results

Here, we provide additional details and material regarding the results of the numerical experiments.

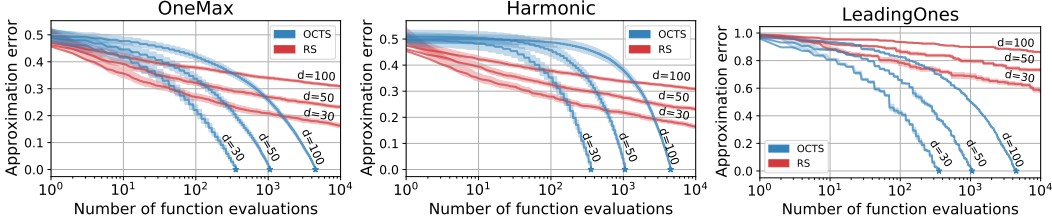

Figure 6: Approximation error in terms of number of function evaluations displayed in log-scale. For each function, we ran the algorithms on various dimensions $d = 30, 50$ and $d = 100$. The star represents the iteration where the algorithm identifies the global optimum.

**Convergence rates.** For the first set of experiments (Figure 6), we ran both RS and OCTS (using the tree provided in Figure 1 with a root node chosen at random). We ran each algorithm 10 times using a budget of $10^4$ evaluations on each function of the benchmark (OneMax, Harmonic and LeadingOnes). For each run, we recorded the value of the approximation error $\max_{x \in \mathcal{X}} f(x) - \max_{i=1...t} f(x_i)$ in terms of number of function evaluations $1 \leq t \leq 10^4$. The results are displayed in Figure 6. We make the following observations:

1. In terms of convergence rates, regardless of the test problem and dimensionality, RS achieves a decreasing rate of order $O(-_l og(n))$ (as indicated by Proposition B.2), while OCTS can achieve an exponentially faster rate of order $O(-n)$ after a certain threshold (as indicated by Theorem 5.3). It validates the results obtained in the theoretical analysis.

2. In terms of number of function evaluations, OCTS successfully manages to find the global optimum of all the problems with a budget $n < 10^4$, while it is unrealistic to identify it using baseline strategies such as Random Search when the budget is $n < 10^4$ whenever $d > 30$.

**Scaling.** For the second set of experiments (Figure F.4), we computed the expected number of iterations required to identify the global optimum for OneMax, Harmonic, LeadingOnes. We used both OCTS and RS. The setup was tested using dimensionalities $d$ ranging from 2 to 100. For each dimensionality, we ran OCTS until it identified the global optimum. We recorded the number of iterations required to identify the global optimum:

$$N^* = \min\{1 \leq t \geq 1 : f(x_t) = \max_{x \in \mathcal{X}} f(x)\}$$

for each test function. Note that since identifying the global optimum is not realistic for the Random Search (e.g. $N^* > 10^{15}$ when $d > 60$ which is unrealistic for a numerical estimation), we used the theoretical result of Proposition B.4 to get the expected number of iterations required to identify the global optimum.

1. First, it is interesting to note that to identify the global optimum using a baseline such as Random Search, it requires at least $\Omega(2^d)$ function evaluations. Thus, it is unrealistic to optimize systems using baselines as soon as $d > 30$.

2. On the other hand, we can identify the global maximum in $O(d^2 \log(d))$ iterations using OCTS. The message here is that it is still possible to identify the global maximum of black-box combinatorial functions using a reasonable evaluation budget of $O(d^2 \log(d))$ function evaluations. This suggests that in practice, with a budget of $n$ evaluations we can expect to successfully optimize systems with a dimensionality up to $d = n^{1/3}$, which allows to cover a wider set of problems that could not be optimized previously.

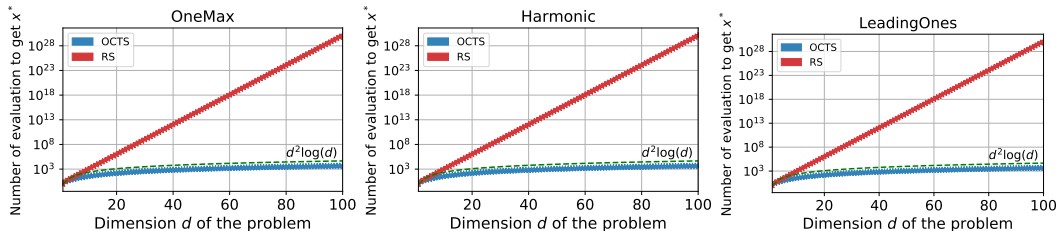

Figure 7: Expected number of iterations required to identify the global optimum of the test problems OneMax, Harmonic and LeadingOnes (in log-scale) in terms of dimensionality $d$ of the problem.

**Choice of the tree.** Here, we investigate the impact of the choice of the tree over the performance of the OCTS algorithm. To do so, we performed 10 runs of OCTS with various trees on the problem presented in the experimental section. The results are reported in the table below.

| | Ising (20) | CT (20) | LABS (20) | MIS (20) | Ising (50) | CT (50) | LABS (50) | MIS (50) |
|---|---|---|---|---|---|---|---|---|
| $T + R$ | 20 (00) | 4.00 (00) | 7.33 (0.88) | 10 (00) | 50 (00) | 10 (00) | 5.17 (0.33) | 18 (1.0) |
| $T + R^*$ | 20 (00) | 4.00 (00) | 7.00 (0.88) | 10 (00) | 50 (00) | 10 (00) | 5.38 (0.46) | 21.2 (1.6) |
| $\pi(T) + R$ | 20 (00) | 3.80 (0.12) | 6.97 (0.72) | 10 (00) | 45.2 (2.71) | 8.76 (0.08) | 5.19 (0.22) | 19.2 (2.0) |
| $\pi(T) + R^*$ | 20 (00) | 3.84 (0.32) | 6.24(0.62) | 10 (00) | 46.4 (1.49) | 8.55 (0.25) | 5.26 (0.32) | 20.6 (1.01) |
| $\pi^*(T) + R$ | 20 (00) | 3.8 (0.0) | 5.88(0.00) | 10 (00) | 50 (00) | 9.40 (00) | 4.32 (00) | 25.2 (0.1) |

Table 2: Performance of the OCTS algorithm with different trees set as input

The table reports the best value observed after $n = 10 * d^2$ evaluations (with standard deviation) where $T$ denotes the tree of Figure 1 with $x_{l,i} = \text{Bin}_l(i) + \vec{0}_{d-l}$. $R$ denotes a root node sampled uniformly over the input space $\{0,1\}^d$, $\pi(T)$ denotes a random permutation of the order of variables. Finally, $R^*$ set the root as the best points obtained from a RS with budget of $d$ evaluations. $\Pi^*(T)$ denotes the ordering where the variables in the tree are ranked according to the best function values recorded by switching the bit corresponding to the given variable and recording it ($d$ evaluations in total to get this ordering). As it can be seen, on most test problems OCTS is robust to the choice of the root node in the sense that for a randomly chosen root (line T+R), the algorithm consistently finds similar optima with low std. Moreover, it is interesting to note that using random permutation ($\pi(T) + R$) does not improve the stability of the algorithm, which is due to the fact that on some problems (e.g. LABS and CT) there is a sequential link between the variables which is preserved by using $T$ and not when permuting the variables. Thus, in practice it is recommended to keep the natural ordering of the variables.

**Comparison with Bayesian optimization.** To motivate the choice of not including Bayesian optimization in the main experiments of the paper, we provide here an additional experiment. Recall first that, according to Table 1 presented in the subsection computational time, Bayesian methods take approximately 1 minute to query a novel point. Thus, since it takes 1 hour for the current Bayesian method to perform 100 function evaluations, it might be even faster to perform an exhaustive search than querying a single point in some cases (whenever the function is cheap or moderate to

evaluate). However, to have an idea of a simple comparison of the proposed algorithms with Bayesian optimization, we performed the following experiment: we took the algorithms of the paper with the budget of the experiments set to $100 * d^2$ evaluations (as set in the paper) and compared it to Bayesian optimization with a budget of 100 evaluations on the contamination problems with different $\lambda \in \{0.00, 0.01, 0.0001\}$. The table below reports the number of evaluations done by each methods, as well as the clock time taken by each method to perform this number of iteration and the best value for the maximum observed so far.

| | OCTS | GA | EA | RS | RLS | GHC | SA | Bayesian |
|---|---|---|---|---|---|---|---|---|
| # evals | 62500 | 62500 | 62500 | 62500 | 62500 | 62500 | 62500 | 100 |
| Conta (0.00) | -21.35 (32s) | -21.42 (43s) | -21.35 (28s) | -21.57 (24s) | -21.52 (24s) | -21.61 (21s) | -21.43 (23s) | -21.57 (88min) |
| Conta (0.01) | -21.52 (34s) | -21.60 (45s) | -21.56 (30s) | -21.73 (23s) | -21.68 (22s) | -21.78 (22s) | -21.78 (22s) | -21.74 (91min) |
| Conta (0.001) | -21.35 (35s) | -21.45 (45s) | -21.36 (25s) | -21.57 (23s) | -21.49 (23s) | -21.61 (21s) | -21.45 (23s) | -21.72 (87min) |

Table 3: Comparison of the algorithms on the Contamination problems with different $\lambda \in \{0, 0.01, 0.0001\}$

As it can be seen, for cheap-to-evaluate black-box system, we can only query 100 points in more than an hour with Bayesian methods while with other methods we can easily query 50000 points in less than a minute. Of course, it results in a less competitive algorithm with such a different available budget (higher is better). However, we point out that Bayesian optimization is suitable for problems where the cost of evaluating the black-box evaluation is significantly larger than the time to generate the next sample point, and we strongly advise using them in this case.

**Real-world problems.** For this set of experiments, OCTS was compared with six different methods commonly used to solve combinatorial black-box problems: Simulated annealing (SA), Random Search (RS), Randomized Local Search (RLS), Genetic Algorithm (GA), Evolutionary Algorithm (EA) and Greedy Hill-Climbing (GHC) described in the previous section. The algorithms were compared using the standardized IOH benchmark of [15] and the test problems provided in [39]: LABS, Concatenated Trap, MaxSAT, MIS, Ising and Contamination. For the problems with a free dimensionality (LABS, Concatenated Trap, MIS, Ising), we tested the algorithms on various dimensions $d \in \{20, 30, 50, 70\}$ to have a broader analysis of OCTS. For MaxSAT, the dimensionality is imposed to be $d = 28, 43$ and $60$. For the contamination problem, the dimensionality is fixed to $d = 25$ but following [39], we tested the algorithm with $= 0.0, 0.01, 0.0001$. To ensure fairness, we performed 10 different runs of each algorithm with different seeds for each algorithm using the same random initialization point $x_1$ when possible (e.g., EA, GHC, SA, RLS) and for the OCTS we used the tree of Section 3 with the root node set to the same initialization point. Finally, we recorded the value of the best value observed so far $\max_{i=1...t} f(x_i)$ at each iteration $t \geq 1$. Results are collected in Figures 8, 9, 10, 11, 12 and 13. We make the following remarks:

- LABS. On each dimension, OCTS is able to identify sequences with scores that could not be reach with any other methods. Moreover, OCTS is constantly the fastest algorithm on this problem for any dimension. Finally, it is interesting to note that the standard deviation tends to descrease as the dimensionality grows.

- Concatenated trap. On each dimension, OCTS is able to identify sequences with scores that cannot be reach with any other method. Again, it is also the fastest algorithm of the benchmark on any dimension of this problem.

- MIS. This is the problem of the benchmark where OCTS struggles the most. However, it is interesting to note that on each dimension it almost finds the best function values with the allocated budget, and still remains competitive. More precisely, on dimension 20, it still finds the best value among all algorithms and is among the fastest on dimensions 50 and 70. However, when $d = 30$, we suspect the problem presents a low conditioning (since RS is almost as efficient as other methods on MIS 30).

- MaxSAT. Again, for dimensions 28 and 43, OCTS is able to identify novel optima that none of the other method could identify. Moreover, it is also the fastest algorithm of the benchmark. On MaxSAT, it successfully identifies the best optimum in par with SA.

- Ising. on this problem, OCTS successfully identifies the global optimum in par with SA. It is interesting to note that it is the fastest algorithm to identify the best optimum.

- Contamination. Again, on this problem OCTS is able to identify novel optima that and is the fastest algorithm of the benchmark.

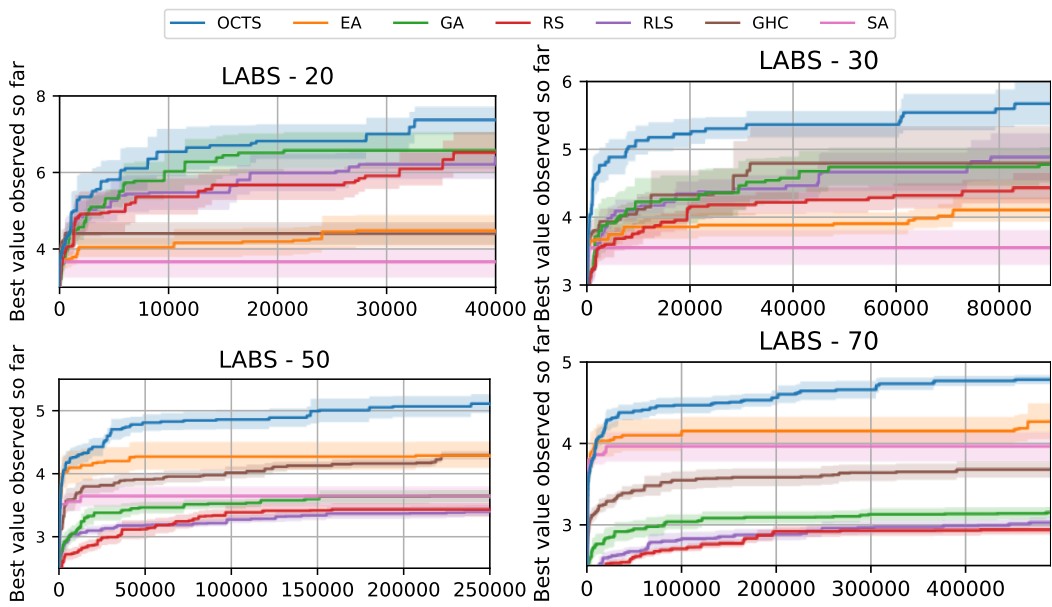

Figure 8: Results of the numerical experiments on the LABS problem. Best valued observed so far in terms of number of function evaluations over ten runs with ±0.5 standard deviation in transparent. Format: Problem name - Dimension.

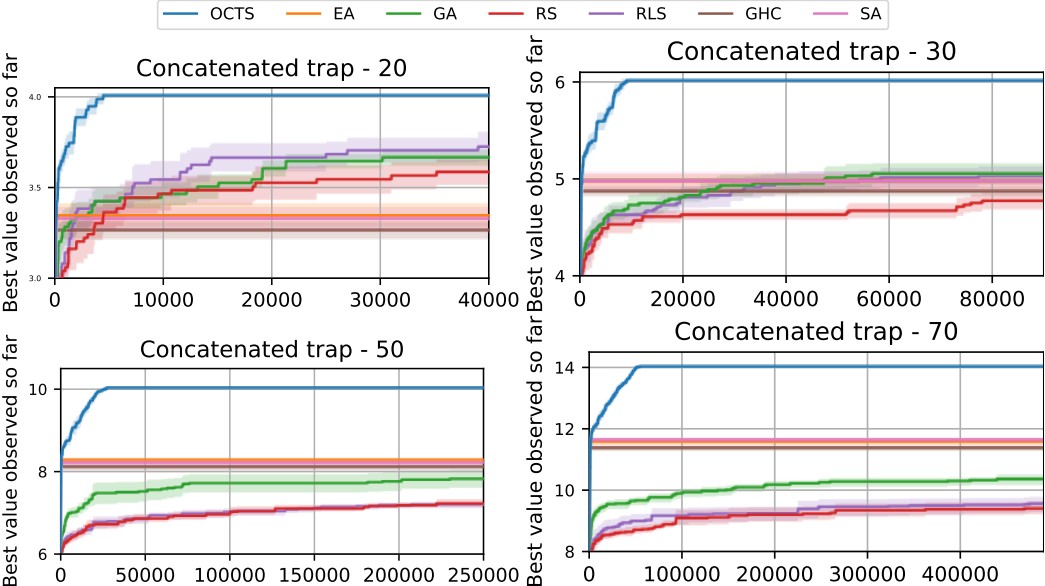

Figure 9: Results of the numerical experiments on the Concatenated trap problem. Best valued observed so far in terms of number of function evaluations over ten runs with ±0.5 standard deviation in transparent. Format: Problem name - Dimension.

Overall, these experiments show that OCTS outperforms existing methods on a wide set of problems. More importantly, this set of experiments shows that using tree search strategies allows to identify novel optima that could not be reach with other methods on some problems (e.g. LABS, Concatenated Trap, MaxSAT and Contamination).

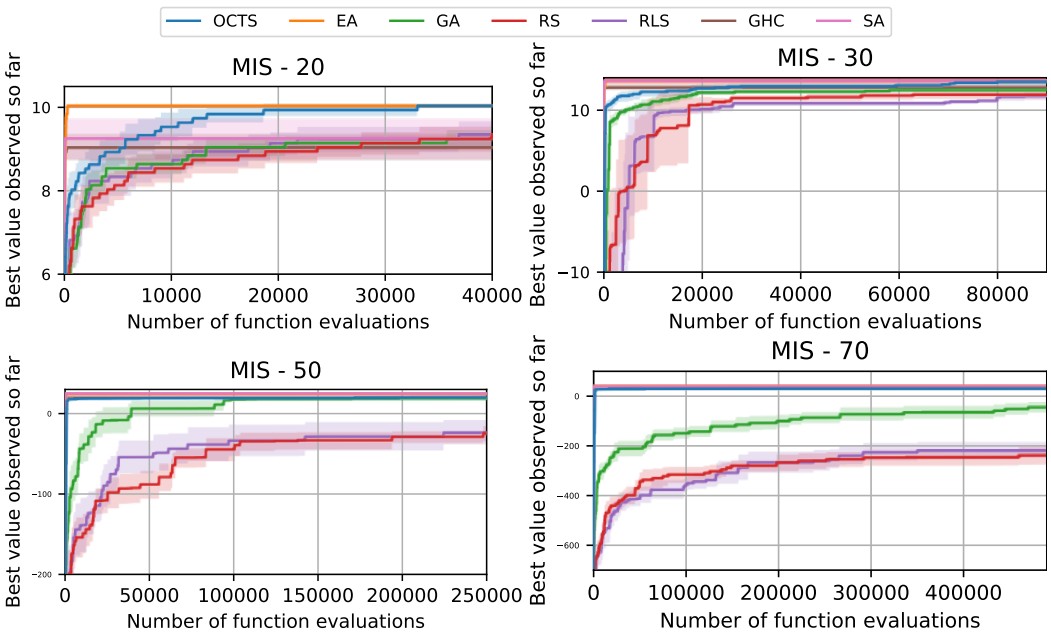

Figure 10: Results of the numerical experiments on the MIS problem. Best valued observed so far in terms of number of function evaluations over ten runs with ±0.5 standard deviation in transparent. Format: Problem name - Dimension.

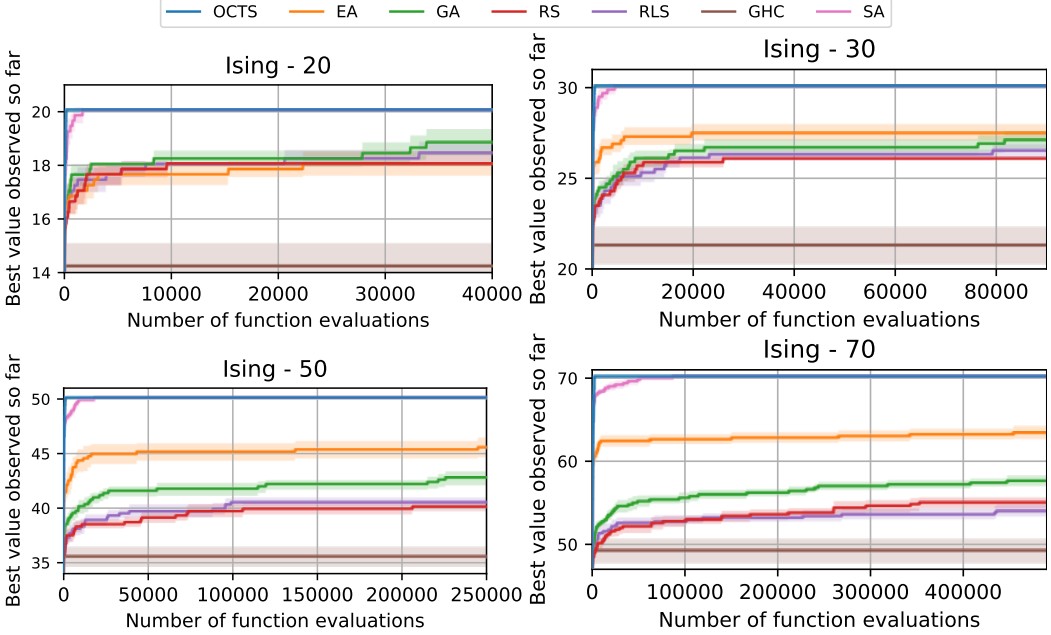

Figure 11: Results of the numerical experiments on the Ising problem. Best valued observed so far in terms of number of function evaluations over ten runs with ±0.5 standard deviation in transparent. Format: Problem name - Dimension.

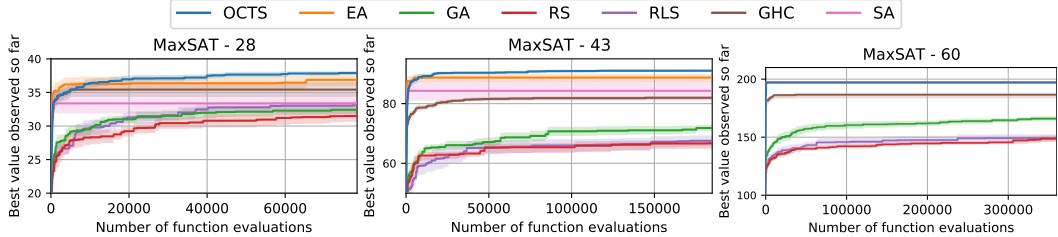

Figure 12: Results of the experiments on the MaxSAT problem. The plot display the best value observed so far in terms of number of function evaluations.

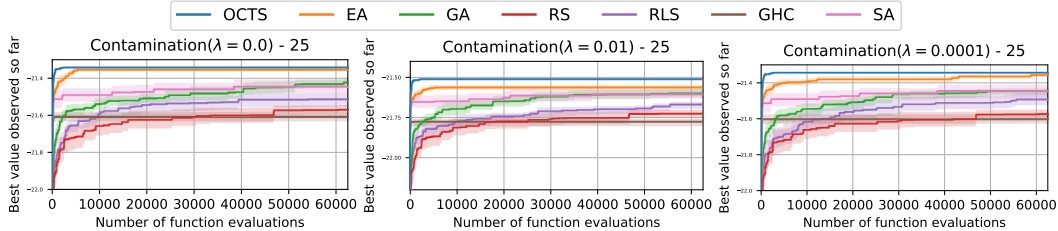

Figure 13: Results of the experiments on the Contamination problem. The plot display the best value observed so far in terms of number of function evaluations.

# G  Proofs of the theoretical results

In this section, we provide the proofs of the theoretical results of the paper.

**Proof of Proposition B.1.** Consider any deterministic algorithm $A$, any $n < |X|$ and any $C > 0$ arbitrarily large. Then, let $f_0(x) = 0$ for all $x \in \mathcal{X}$ be the zero function and let $x_1 \ldots, x_n$ denotes the series of evaluation points generated by the algorithm $A$ on $f_0$. Then, if we consider the function $f_{A,C}(x) = C\mathbb{I}\{x \notin \{x_1, \ldots, x_n\}\}$, the algorithm $A$ will generate the same evaluations points $x_1, \ldots, x_n$ as the one generated on $f_0$ since they share the same function values over these points. Therefore, we will have $\max_{i=1\ldots n} f_{A,C}(x_i) = 0$ and since $\max_{x \in \mathcal{X}} f_{A,C}(x) = C$, we deduce that $\max_{x \in \mathcal{X}} f_{A,C}(x) - \max_{i=1\ldots n} f_{A,C}(x_i) \geq C$ which concludes the proof.

**Proof of Proposition B.2.** Consider any $f \in \mathrm{Lip}(k)$, set any $n \geq 1$ and consider the tree representation $T$ defined in Section 3. By definition of the tree and since the sequential search is a breadth-first search, we know that after $n$ iterations, all the points on any level $l$ with $2^l \leq n$ of the tree will have been evaluated. Moreover, by definition of the tree, the level $l$ contains all the different combination of points on the first $l$ components. Thus, if we denote by $x^\star \in \arg\max_{x \in \mathcal{X}} f(x)$ any maximiser of the objective function, then there exists $x^l \in \bigcup_{i=0\ldots 2^l-1} x_{l,i}$ that share the same first $l$ elements as $x^*$ and therefore satisfies:

$$d_H(x^l, x^*) = \sum_{i=1}^{d} \mathbb{I}\{x_i^l \neq x_i^*\} = \underbrace{\sum_{i=1}^{l} \mathbb{I}\{x_i^l \neq x_i^*\}}_{=0} + \sum_{i=l+1}^{d} \mathbb{I}\{x_i^l \neq x_i^*\} \leq (d-l).$$

Now, since the function is $k$-Lipschitz and $x^l$ has been evaluated, we have:

$$\max_{x \in \mathcal{X}} f(x) - \max_{i=1\ldots n} f(x) \leq f(x^*) - f(x^l) \leq k d_H(x^*, x^l) \leq k(d-l).$$

Finaly, since the previous inequality holds for any $l$ such that $n \geq 2^l$, we directly get the result by setting $l = \lfloor \ln(n)/\ln(2) \rfloor$.

**Proof of Proposition B.3.** Consider any $f \in \mathrm{Lip}(k)$, any $n \geq 2^{d/2}$ and set $L = d - \lfloor \ln(n)/\ln(2) \rfloor$. Denote by $\mathcal{X}_{kL} := \{x \in \mathcal{X} : f(x) \geq \max_{x \in \mathcal{X}} f(x) - k \times L\}$ the level set of the function corresponding to the desired accuracy and observe that since $f \in \mathrm{Lip}(k)$ then $B(x^*, L) = \{x \in \mathcal{X} : d_H(x, x^*) \leq L\} \subseteq \mathcal{X}_{kL}$ for any global maximiser $x^* \in \arg\max_{x \in \mathcal{X}} f(x)$. Thus, since $L$ has been chosen so that $d/L \geq 2$, using the standard bounds on binomial coefficients $\binom{d}{L} \geq (d/L)^L$, we have the following:

$$\frac{|B(x^*, L)|}{|\mathcal{X}|} = \frac{\sum_{i=0}^{L} \binom{d}{i}}{2^d} \geq \frac{\binom{d}{L}}{2^d} \geq \frac{(d/L)^L}{2^d} \geq 2^{L-d} \geq \frac{1}{n}.$$

Therefore, it follows that:

$$\begin{aligned}
\mathbb{P}\left(\max_{x \in \mathcal{X}} f(x) - \max_{i=1\ldots n} f(x_i) \leq kL\right) &= \mathbb{P}\left(\bigcup_{i=1}^{n}\{x_i \in \mathcal{X}_L\}\right) \\
&= 1 - \mathbb{P}\left(x_i \notin \mathcal{X}_L\right)^n \\
&\geq 1 - \mathbb{P}\left(x_i \notin B(x^\star, L)\right)^n \\
&= 1 - \left(1 - \frac{|B(x^*, L)|}{|\mathcal{X}|}\right)^n \\
&\geq 1 - \left(1 - \frac{1}{n}\right)^n \\
&\geq 1 - \frac{1}{e}
\end{aligned} \tag{5}$$

which proves the result, where we used on the first inequality the fact that $B(x^*, L) \in \mathcal{X}_{kL}$ and on the third the inequality $1 + x \leq e^x$.

**Proof of Proposition B.4.** Let $x_1, x_2, x_3, \ldots$ be a series of evaluation points independently and uniformly distributed over $\mathcal{X} = \{0, 1\}^d$ and let $\mathcal{X}^* = \arg\max_{x \in \mathcal{X}} f(x)$ denotes the set of global

optimum. First, observe that for any $t \geq 1$, $\mathbb{P}(x_t \in \arg\max_{x \in \mathcal{X}} f(x)) := p = |\mathcal{X}^*|/|\mathcal{X}|$. Then, we have that

$$
\begin{aligned}
\mathbb{E}[N^*] &= \sum_{t=1}^{\infty} t \times \mathbb{P}(N^* = t) \\
&= \sum_{t=1}^{\infty} t \times \mathbb{P}(\{x_t \in \mathcal{X}^*\} \cap \{x_{t'} \notin \mathcal{X}^*\}_{t'=1}^{t-1}) \\
&= \sum_{t=1}^{\infty} t \times p(1-p)^{t-1} \\
&= \frac{p}{(1-p)} \sum_{t=1}^{\infty} t(1-p)^t \\
&= \frac{p}{1-p} \times \frac{(1-p)}{p^2} \\
&= \frac{|\mathcal{X}|}{|\mathcal{X}^*|}
\end{aligned}
\tag{6}
$$

which concludes the proof.

**Proof of Proposition D.1.** We start to bound the number $n^*$ of iterations required to identify a global optimizer of the objective function $f$. Let $N = \sum_{l=0}^{d-1} |I_l| + 1$ be the upper bound on $n^*$ and let $(l_2, i_2), \ldots, (l_t, i_t)$ be the index of the nodes $(l_t, i_t)$ selected by the algorithm (line 4) at each iteration $t \geq 2$. Consider two cases: (1) $n^* \leq N - 1$ and (2) $n^* > N - 1$. First, observe that the result trivially holds if $n^* \leq N - 1$. Now assume that $n^* > N - 1$ which necessarily implies $(i_{n^*}, l_{n^*}) \notin \{(l_t, i_t)\}_{t=1}^{N-1}$. Since we know that the algorithm only selects nodes in $\bigcup_{l=0}^{d-1} I_l$ until it identifies the global optimum, we know that $(l_t, i_t) \in \bigcup_{l=0}^{d-1} I_l$ for all $2 \leq t \leq N - 1$. Moreover, since the algorithm selects a different node at each iteration, we have $(l_t, i_t) \notin \{(l_{t'}, i_{t'})\}_{t' < t}$ for all $2 \leq t \leq N - 1$. Thus, considering iteration $N$, we have:

$$
(l_N, i_N) \in \bigcup_{l=0}^{d-1} I_l / \{(i_t, l_t)\}_{t=2}^{N-1}
$$

Now, since (1) $|\{(i_t, l_t)\}_{t=2}^{N-1}| = N - 2$, (2) $|\bigcup_{l=0}^{d-1} I_l| = N - 1$, (3) $(l_{n^*}, i_{n^*}) \notin \{(i_t, l_t)\}_{t=1}^{N-1}$ and (4) $(l_{n^*}, i_{n^*}) \in \bigcup_{l=0}^{d-1} I_l$, we necessarily have $(l_N, i_N) = (l_{n^*}, i_{n^*})$ which means that the global optimizer is identified at time $N$ and completes the proof.

Now, we prove the finite-time bound by using similar arguments as in [37]. Let $(l_{\max}(n), i_{\max}(n))$ be the index of (one of) the deepest node that has been selected up to round $n$, i.e., $l_{\max}(n) = \max_{t=2\ldots n} l_t$. We prove by contradiction that $l_{\max}(n) \geq l(n)$. Assume that $l_{\max}(n) < l(n)$. Then, it implies that $l_t \leq l(n) - 1$ for all $2 \leq t \leq n$. Moreover, since $n < n^*$, we also know that $(l_t, i_t) \in \bigcup_{l=0}^{d-1} I_l$, which combined with the previous statement gives that $(l_t, i_t) \in \bigcup_{l=0}^{l(n)-1} I_l$ for all $2 \leq t \leq n$. Now since, $n - 1 > \sum_{l=0}^{l(n)-1} |I_l|$ by definition, we obtain the following contradiction:

$$
\sum_{l=0}^{l(n)-1} |I_l| < n - 1 = |\{(l_t, i_t)\}_{t=2}^{n}| \leq \left| \bigcup_{l=1}^{l(n)-1} I_l \right| = \sum_{l=0}^{l(n)-1} |I_l|
$$

which proves that $l_{\max}(n) \geq l(n)$. Now since $l_{\max}(n) \geq l(n)$ and $(l_{\max}(n), i_{\max}(n)) \in \bigcup_{l=0}^{d-1} I_l$ since $n < n^*$, we have that:

$$
\begin{aligned}
\max_{i=1\ldots n} f(x_i) &\geq f(x_{l_{\max}(n), i_{\max}(n)}) \\
&\geq f(x^*) - k\mathrm{Diam}(\mathcal{X}_{l_{\max}(n), i_{\max}(n)}) \\
&\geq f(x^*) - k(d - l_{\max}(n)) \\
&\geq f(x^*) - k(d - l(n))
\end{aligned}
\tag{7}
$$

which concludes the proof.

**Proof of Lemma D.3.** Using the definition of $I_l$ and the fact that (1) $\text{Diam}(\mathcal{X}_{l,i}) = d - l$ and (2) $f(x) \geq f(x^*) - k^* d_H(x, x^*)$, we obtain the following inclusions:

$$
\begin{aligned}
I_l &= \{\text{nodes}(l, i) \in T \text{ s.t. } f(x_{l,i}) + k(d - l) \geq f(x^*)\} \\
&\subseteq \{\text{nodes}(l, i) \in T \text{ s.t. } f(x^*) - k^* d_H(x_{l,i}, x^*) + k(d - l) \geq f(x^*)\} \quad\quad (8) \\
&= \{\text{nodes}(l, i) \in T \text{ s.t. } d_H(x_{l,i}, x^*) \leq k/k^*(d - l)\}
\end{aligned}
$$

Now using the fact that for all $(x, x') \in \mathcal{X}^2$, we have $d_H(x, x') = \sum_{i=1}^{d} \mathbb{I}\{x_i \neq x_i'\} \geq \sum_{i=1}^{l} \mathbb{I}\{x_i \neq x_i'\} := d_l(x, x')$, we obtain that:

$$
\begin{aligned}
I_l &\subseteq \{\text{nodes}(l, i) \in T \text{ s.t. } d_l(x_{l,i}, x^*) \leq k/k'(d - l)\} \\
&= \bigcup_{L=0}^{\lfloor k/k^*(d-l) \rfloor} \{\text{nodes}(l, i) \in T \text{ s.t. } d_l(x_{l,i}, x^*) = L\}.
\end{aligned} \quad\quad (9)
$$

Finally, by definition of the tree, all the nodes at level $l$ contain all the combinations $\{0, 1\}^l$ on their first $l$ elements. Thus, it follows that:

$$
\begin{aligned}
|I_l| &\leq \sum_{L=0}^{\lfloor k/k^*(d-l) \rfloor} |\{\text{nodes}(l, i) \in T \text{ s.t. } d_l(x_{l,i}, x^*) = L\}| \\
&= \sum_{i=0}^{\lfloor k/k^*(d-l) \rfloor} \binom{l}{i}
\end{aligned} \quad\quad (10)
$$

which proves the result.

**Proof of Lemma D.4.** First, note that the condition $l \geq 2cd/(1 + 2c)$ implies that $c(d - l) \leq l/2$. Then, using the bound of [12] using the binary cross entropy $B(x) = -x \log(x) - (1 - x) \log(1 - x)$, we have that

$$
\sum_{i=0}^{\lfloor c(d-l) \rfloor} \binom{l}{i} \leq 2^{B(c(d-l)/l)l}.
$$

Now setting $y = c(d - l)/l$ and observing that $l/d = c/(c + y)$, we have

$$
B(c(d - l)/l)d = B(c(l - d)/l) \times \frac{l}{d} \times d = \frac{B(y)}{1 + y} \times \frac{1 + y}{c + y} \times c \times d. \quad\quad (11)
$$

Finally, using the fact that $B(y)/(1 + y) \leq 1/2$ and that $(1 + y)/(c + y) \leq (1 + 1/2)/(c + 1/2)$ for all $y \leq 1/2$, we thus obtain that

$$
B(c(d - l)/l)l \leq \left( \frac{3c}{2c + 1} \right) \times \frac{d}{2}
$$

which proves the lemma.

**Proof of Theorem 4.3.** By virtue of Proposition D.1, recall that $\max_{x \in \mathcal{X}} f(x) - \max_{i=1...n} f(x_i) \leq k(d - (n))$ where $l(n) = \min\{0 \leq L \leq d - 1 : \sum_{l=1}^{L} |I_l| \geq n - 1\}$. So first, observe that since $|I_l| \leq 2^l$ for all $0 \leq l \leq d - 1$, it necessarily follows that $\sum_{l=0}^{L} |I_l| \leq 2^{L+1} - 1$. Thus, independently of the conditioning number $c$, we have $l(n) \geq \lceil \ln(n)/\ln(2) \rceil - 1$ which proves the first part of the result. To prove the second part of the result, we need an intermediate lemma to bound the the size $|I_l|$ using Lemma D.4. Now, setting $L_s = \lceil \frac{2c}{1+2c}d \rceil - 1$ and $C = 2^{\left( \frac{3c}{2c+1} \right) \frac{d}{2}}$ and using the previous bound, we have for all $L > L_s$:

$$
\sum_{l=0}^{L} |I_l| \leq \sum_{l=0}^{L_s} 2^l + \sum_{l=L_s+1}^{L} C = 2^{L_s+1} - 1 + C(L - L_s)
$$

Thus, we deduce that whenever $n > n_c = 2^{L_s+1} = 2^{\lceil \frac{2c}{1+2c}d \rceil}$, we have $l(n) \geq \lceil L_s + \frac{n-n_c}{C} \rceil$ which concludes the proof by observing that $L_s = \frac{\ln(n_c)}{\ln(2)} - 1$.

**Proof of Proposition 5.2.** The proof is based on the arguments of [37]. Let $x_{l^*, i_l^*}$ be a node of tree that contains (one of) the global optimum with the deepest level $l^* \in \{0, \dots, d\}$ and let $i_l^*$ be the index of the node$(l, i_l^*)$ at level $0 \le l < l^*$ which contains $x_{l^*, i_l^*}$ as a child. Now, let $\tau_l = \min\{1 \le t \le 2^d : f(x_{l, i_l^*})$ is expanded$\}$ be the time where the node of level $l < l^*$ containing $x_{l^*, i_l^*}$ is expanded and let $l_t^* = \max\{0 \le l \le d - 1 : x_{l, i_l^*}$ has been expanded before $t\}$ be the level of the deepest expanded node containing $x_{l^*, i_l^*}$ at time $1 \le t \le n$. First, if $l_n^* = l^* - 1$, then the result trivially holds. Now assume that $l_n^* < l^* - 1$ and observe that we have the property that any node of level $l + 1$ is expanded at a time $t \in \Delta_l := [\tau_l, \tau_{l+1} - 1]$ belongs to $I_{l+1}^*$ for any $l \le l^* - 2$. Indeed, a node$(l+1, i)$ is expanded during $\Delta_l$ if and only if $f(x_{l+1, i}) \ge f(x_{l+1, i_{l+1}^*})$ since $f(x_{l+1, i_{l+1}^*})$ has already been evaluated and thus $f(x_{l+1, i}) + k_{\min}\text{Diam}(\mathcal{X}_{l+1, i}) \ge f(x_{l+1, i_{l+1}^*}) + k_{\min}\text{Diam}(\mathcal{X}_{l+1, i_{l+1}^*}) \ge f(x^*)$ which means that $(l+1, i) \in I_{l+1}^*$. However, it is also possible that during a round of optimal nodes selection, no node of level $l+1$ is expanded because their is a node$(j, i)$ with a lower level $0 < j < l + 1$ has a higher value. Nonetheless, in that case we have $(j, i) \in I_j^*$ since

$$f(x_{j,i}) \ge f(x_{l+1, i_{l+1}^*}) \ge f(x^*) - k_{\min}\text{Diam}(\mathcal{X}_{l+1, i_{l+1}^*}) \ge f(x^*) - k_{\min}\text{Diam}(\mathcal{X}_{j,i}).$$

Thus, since each batch of selection of potentially optimal nodes result in at most $d$ expansions, we deduce that for any $0 \le l \le l^* - 2$,

$$\tau_{l+1} - \tau_l \le \left[ \sum_{(l+1,i) \in I_{l+1}^*} \mathbb{I}\{(l+1, i) \text{ is expended during } \Delta_l\} + \sum_{h=1}^{l} \sum_{(h,i) \in I_h^*} \mathbb{I}\{(h, i) \text{ is expended during } \Delta_l\} \right] d$$

$$= d \sum_{h=1}^{l+1} \sum_{(h,i) \in I_h^*} \mathbb{I}\{(h, i) \text{ is expended during } \Delta_l\} \tag{12}$$

Thus, since $l_n^* \le l^* - 2$, we have

$$\sum_{l=0}^{l_n^*} \tau_{l+1} - \tau_l \le d \sum_{l=0}^{l_n^*} \sum_{h=1}^{l+1} \sum_{(h,i) \in I_h^*} \mathbb{I}\{(h, i) \text{ is expended during } \Delta_l\}$$

$$= d \sum_{h=1}^{l_n^*+1} \sum_{l=h-1}^{l_n^*} \sum_{(h,i) \in I_h^*} \mathbb{I}\{(h, i) \text{ is expended during } \Delta_l\} \tag{13}$$

$$\le d \sum_{h=1}^{l_n^*+1} \sum_{(h,i) \in I_h^*} \mathbb{I}\{(h, i) \text{ is expended at any time}\}$$

$$\le d \sum_{h=1}^{l_n^*+1} |I_h^*|$$

Finally, since $\tau_{l_n^*+1} > n$ by definition and $\tau_0 = 1$, we deduce that

$$n < 1 + d \sum_{l=1}^{h_n^*+1} |I_l^*| \le d \sum_{l=0}^{h_n^*+1} |I_l^*|$$

Then, for any $l(n)$ such that $n \ge d \sum_{l=1}^{l(n)} |I_l^*|$, we have $l_n^* \ge \min(l^*, l(n))$. Finally, since by definition the node$(l_n^*, i_n^*)$ has been expanded, we have:

$$\max_{x \in \mathcal{X}} f(x) - \max_{i=1 \dots n} f(x_i) \le f(x_{l^*, i_{l^*}^*}) - f(x_{l_n^*+1, i_{l_n+1}^*})$$

$$\le k_{\min} d_H(x_{l^*, i_{l^*}^*}, x_{l_n^*+1, i_{l_n+1}^*}) \tag{14}$$

$$\le k_{\min}(d - l(n) - 1)$$

which concludes the proof.

**Proof of Theorem 5.3.** Recall that by virtue of Proposition 5.2, we have $\max_{x \in \mathcal{X}} f(x) - \max_{i=1 \dots n} f(x_i) \le k_{\min}(d - l(n) - 1)$ with $l(n) = \max\{0 \le L \le d - 1 : (n/d) \ge \sum_{l=0}^{L} |I_l^*|\}$.

It just remains to bound the terms $|I_l^*|$. Similarly to the proof of Theorem 4.3, we have $|I_l^*| \leq 2^l$ for all $0 \leq l \leq d-1$ independently of the value of $c$. Therefore, it follows that $\sum_{l=0}^L |I_l^*| \leq 2^{L+1} - 1$ which ensures that $l(n) \geq \left\lfloor \frac{\ln(n/d+1)}{\ln(2)} \right\rfloor - 1 \geq \left\lfloor \frac{\ln(n/d)}{\ln(2)} \right\rfloor - 1$ and proves the first part of the result.

For the second part of the result, we follow the proof of Theorem 4.3. Setting $L_s = \left\lceil \frac{2c}{1+2c} d \right\rceil - 1$ and $C = 2^{\left(\frac{3c}{2c+1}\right)\frac{d}{2}}$ and using Lemma D.4, gives that for all $L > L_s$:

$$\sum_{l=0}^L |I_l| \leq \sum_{l=0}^{L_s} 2^l + \sum_{l=L_s+1}^L C = 2^{L_s+1} - 1 + C(L - L_s).$$

Therefore, it follows that for all $n > n_c = d2^{L_s+1} \geq d(2^{L_s+1} - 1)$, we have $l(n) \geq L_s + \left\lfloor (n/d + 1 - 2^{L_s+1})/C \right\rfloor$ and we deduce that:

$$d - l(n) - 1 \leq d - \left\lfloor \frac{n/d - 2^{L_s+1}}{C} \right\rfloor - L_s - 1 \leq d - \left\lfloor \frac{n - n_c}{dC} + \frac{\ln(n_c/d)}{\ln(2)} \right\rfloor$$

by plugging the values of $n_c$ and $L_s$, which proves the second part of the result combined with Proposition 5.2.