# OpenReview forum: "Optimistic Tree Searches for Combinatorial Black-Box Optimization"
_NeurIPS.cc/2022/Conference — NeurIPS 2022 Accept_

### Official Review · Reviewer_Luwu · 2022-06-30

**Rating:** 6
**Confidence:** 3
**Soundness:** 3 good
**Presentation:** 3 good
**Contribution:** 3 good

**Summary:**

This paper addresses the problem of combinatorial black-box optimization. The solution is built upon a tree-structure search procedure with optimistic search strategy. The contribution of the paper in my opinion is two-fold: 1) Algorithmically, it designs a new combinatorial black-box optimization solver OLTS (and its practical variant, OCTS) by adapting the optimistic strategy applied on tree-search optimizer. 2) Theoretically, it provides convergence analysis on the proposed solver (and its variant OCTS) which is shown to be superior than random search.

**Questions:**

Please see some questions in the detailed comments above. Below are some other questions. I would not expect the authors to answer all of them but I am listing them here nonetheless.

1) Is the tree representation of the combinatorial space first proposed in this paper, or was it proposed by some existing papers but perhaps in a different setting and goal?

2) Why is Bayesian optimization not compared as a baseline?

3) The definition of l(n) is a little bit involved, so that it makes it hard to understand how well bounded the solution of OLTS/OCTS would be. Could you illustrate a bit more about the intuition of l(n)? Perhaps using a concrete example to show what is a typical value of l(n).

4) Similar question to n_c in Theorems 4.3 and 5.3: what is the intuition of n_c, and what is a typical value for it? This is important because if n_c is very large (which seems to be from its exponential form), then Theorems 4.3 and 5.3 would be less informative.

5) In the empirical results, the MIS-30 setting yields no good strategy, but in the MIS-70 setting, OCTS and a few other methods achieve optimal performance in the first few iterations. Why does dimension make the structure of the problem so different so that the difficulty of finding a good solution is totally different?




**Limitations:**

The authors claimed that they discussed potential limitations (1.b) and negative societal impacts of the work (1.c) in the Appendix, but I cannot see any obvious discussions of this kind.

**Strengths And Weaknesses:**

Strengths:
1) The structure of the paper is clear and the paper is overall well-written. The clarity is in general good, except for a few points that will be discussed in the weakness part.

2) The problem of combinatorial black-box optimization is an important problem that has vast applicability of various domains, including machine learning.

3) The paper provides the first finite-time linear convergence rates for the problem. It is a significant improvement compared to the logarithmic rates of baselines (random search).

4) The empirical results are promising. The algorithm, though simple, has been shown to be outperforming the baselines on a set of benchmark black-box combinatorial optimization problems, including LABS, MIS, Ising, MaxSAT, and Contamination.


Weakness:
1) The novelty of the proposed solvers, OLTS and OCTS, is limited. Both the tree-based search and the optimistic strategy have been well studied under similar contexts. The main critique from me is not that the algorithms are not novel, but that the novelty is somewhat overclaimed.

For example, the tree based search has been discussed in a few previous papers (e.g., in [39] and also UCT -- UCB for trees). But this has not been acknowledged in the paper. It appears that the tree structure is first proposed in this paper.

As another example, the optimistic strategy for estimating the potential of the tree nodes is also adapted from [39]. Though the paper lists three major differences of OLTS/OCTS vs [39], it still seems incremental. Also, it is not clearly explained why these differences are made to adapt to the tree structure and what are the advantages.

2) It is not clear what are the intuitions of l(n) and n_c in the propositions and theorems, so that it is hard to understand how tight the derived convergence bounds are in the respective theorems/propositions. At least from a first look, the bounds do not seem tight, and therefore the theory is not as informative. The paper would be stronger if these are better explained/clarified.

3) Bayesian optimization is an important category of methods for black-box combinatorial optimization problems, but it is not included in the set of baselines. Why is it? It would be good to explain.

4) The empirical results are promising in general. One question from me, though, is that, what are the reasons that certain problems are selected for evaluation. For example, reference [18] and reference [41] each provided a set of benchmark problems, but this paper selected a subset from each of these two references instead of evaluating all of the settings in either one of them. It does not seem that the proposed OCTS cannot work on the other problems, e.g., the neural architecture search benchmark which is of potential high interest to the ML community.

Minor aspects:
1) The introduction well motivates the paper, but is a bit too condensed. Perhaps better to split it into multiple paragraphs.
2) Line 186: I_h seems to be a typo, it should be I_l
3) Line 270: Proposition A.3 -- is it a typo?

---

> ### Author Response · Authors · 2022-08-02
> **Official Response to Reviewer Luwu**
>
> We thank the reviewer for their valuable feedback. We try to provide a concise answer due to space issues:
>
> 1 First, we strongly apologize if reading the paper gives the feeling of overclaimed contributions which (to be honest) was not done on purpose. Of course, we agree that the optimistic and tree search machinery is not new. In our opinion, the main contributions of the paper are: 1) the introduction of the first combinatorial solvers that use optimistic searches over specific trees, 2) the development of novel theoretical results (which to the best of our knowledge are the first of their kind and bring insightful results for combinatorial problems). and 3) we obtain novel algorithms that display very strong results on benchmarks. In our opinion, it is valuable to the community from both a theoretical and practical side. However, to prevent this feeling of overclaimed contributions, we made the following modifications:
> -  l88, the section "Optimistic Strategies" is now called 'Optimistic Tree Search Strategies' in order to stress that tree based searches have already been discussed in previous works (such as UCB and UCT) and we stress that SOO and DOO use tree searches in continuous spaces
> - l31, "we build upon the works of DIRECT[30] and SOO[41] and show how to use optimistic strategies"
> - l118, 'To implement optimistic tree search strategies [39], we need a hierarchical partition of the combinatorial space'
> - l160, we start with the "OLTS implements the optimistic principle [39] over combinatorial trees". Same for OCTS in l232.
>
> 2 Up to our knowledge, all the works that employ optimistic tree searches only focus on continuous spaces (where one can use a continuous $2^d$-ary tree partition of the space [39]). There is simply no equivalent hierarchical representation for combinatorial spaces reported in the literature (even for different goals). This is why we came up with the non-trivial tree structure of the paper satisfying assumptions 3.1 & 3.3. Although the nature of combinatorial (CombTree) and continuous (ContTree [39]) spaces are different, we list some differences:
> - [width] In CombTee, only one coordinate is switched at each split. Doing the same in ContTree would result in losing the decreasing diameter property (Ass. 3.3). As a consequence, CombTrees only have 2 children per node (independently of the dimension) while in continuous trees we have $2^d$ children per node. Thus, ContTrees are much wider/flat trees that exponentially explode with the dimension. Moreover, since CombTree imposes that left child has the same value as the parent node, an import consequence is that one can easily navigate through the tree linearly with $d$ evaluations while it would require $d*2^d$ evaluations in ContTree which explodes with $d$. Note that this trick imposes that the size of CombTree is  $2^{d+1}$ instead of $2^d$
> - [depth] The depth of CombTree is $d+1$ while the depth of ContTree is infinite. In practice, ContTree  are controlled by a parameter $h$ which impacts the performance. In [38] they obtain two very distinct regimes of convergence that depend on $h$ (exponential and polynomial), while we only obtain a single (fast) linear regime of convergence
> - [storage] the nodes of CombTree can be represented as $x_{l,i}= bin_l(i) + 0_{d-l}$. It allows to simply store the index $(l,i)$ of the tree search instead of the full vectors $x_{l,i}$ of dimension $d$ for a better scaling w.r.t. the dimension
> - [theory] most of the analysis boils down to bounding the volume of the sphere $B(x_{l,i},R)$ for some $R>0$ where $x_{l,i}$ is any point in the tree. In continuous space, it is easy to integrate and proportional to $R^d$, while in combinatorial spaces, the results are discrete (hence $l(n)$) and (combinatorically) explode with $R$. To overcome this phenomenon, we introduce specific combinatorial techniques (see Proofs of Lem B3, Lem B4 and Prop B1)
>
> 3 Informally, $l(n)$ corresponds to the minimum depth at which the tree search will be after $n$ evaluations, which depends on the sets $I_l$ of potential optima. Similarly, $log_2(n_c)$ corresponds to the level after which the number of potential optima per level stops to explode and the algorithm will get a linear behavior. For example, when $f(x) = -d_H(x, \vec{1})$ with the tree of Section 3 and OLTS with $k=1.5$ and $d=4$, we have $|I|_0= 1$, $|I_1|= 2$, $|I_2|=3$, $|I_3|=|I_4|=1$. Thus, in this case we have $l(1)=0$, $l(4)=1$, $l(7)=2$ and $l(8)=3$ and we know that after $8$ iterations we will be at least at the level 3.
> These details are now extended in the main document and examples for the values of $l(n)$ and $n_c$ are illustrated (with a tree) on the previous example in Appendix B with a pointer in the document.
>
> 4 We refer to the resp to rev S1XN for Xps/Bayesian methods/societal Impact and we only considered in the benchmark the problems that are (1) non-synthetic, (2) have a free dimension for [18] and (3) cheap-to-evaluate (taking out NAS)

---

> > ### Comment · Reviewer_Luwu · 2022-08-07
> > **Clarified most concerns**
> >
> > I appreciate that the authors take the time to respond to my comments. I think my concerns -- overclaim of novelty/contributions, baselines (especially Bayesian optimization), benchmark problems, and the intuitions behind the theorems -- are generally clarified by the authors. The paper does contribute to the field of black-box combinatorial optimization regardless of the concerns I have raised. It will be a stronger paper with the issues resolved.

---

> > > ### Author Response · Authors · 2022-08-09
> > > **Final note**
> > >
> > > We thank the reviewer for taking time to read the rebuttal and we are very happy that some of your concerns have been clarified. Finally, we would also like to thank the reviewer for raising its score and for its various remarks which helped to improve the overall quality of the paper. Feel free to point out any additional aspect of our work that could further be improved or clarified.

---

### Official Review · Reviewer_n1bW · 2022-07-11

**Rating:** 4
**Confidence:** 4
**Soundness:** 2 fair
**Presentation:** 2 fair
**Contribution:** 2 fair

**Summary:**

The paper considers the black-box optimization of combinatorial binary functions. The functions are assumed to obey a Lipschitz  condition given some metric on the hypercube. For the optimization problem, the authors propose two algorithms, depending on the knowledge of the Lipschitz constant. Both algorithms rely on tree search and optimistic upper bounds. Theoretical guarantees are provided for the convergence of the algorithms. The empirical work show that the algorithm with unknown Lipschitz constant (OCTS) outperforms the considered baselines on a variety of problems.

**Questions:**

Algorithm 3 (referenced from Algorithm 2) is only in the supplementary material. This poses some problems in the readability of the paper that should be resolved.

**Limitations:**

It is not clear how meaningful the Lipschitz condition is for the practical problems considered (beyond the constant that results from the discrete nature of the problem).

**Strengths And Weaknesses:**

The proposed algorithm is fairly natural given the Lipschitz assumption. The case of unknown Lipschitz constant is treated in a similar way as the DIRECT algorithm (although it is not referenced).

The theoretical results are straightforward, but nevertheless useful.

The binary tree is assumed as provided, but I would assume that the ordering of the indices might have significant influence on the performance.

Given the optimistic tree search approach, the problem is somewhat related to the combinatorial bandit problem. The main difference here is that the function is deterministic, which allows much stronger bounds, but I would assume some techniques from combinatorial bandits could carry over.

The empirical performance is a strong argument for the paper. The baselines are difficult to evaluate, since there is little detail provided regarding their implementation and parametrization.

---

> ### Author Response · Authors · 2022-08-02
> **Official Response to Reviewer n1bW**
>
> First, we would like to thank the reviewer for its feedback. We try to address the remarks below:
>
> 1. The DIRECT algorithm is first cited in the introduction of the document in l31-32 where we introduce the contribution: "More precisely, we build upon the works of [30, 39] and show how to use optimistic tree searches on combinatorial spaces". To the best of our knowledge, [30] is the original paper of the DIRECT algorithm. However, to make the connection more clear, we now use "the DIRECT[30] and SOO[39] algorithms" and add on top of Definition 5.1 "we consider a set of potentially optimal nodes similar to the one of the DIRECT algorithm [30]"
>
> 2. To the best of our knowledge, it is the first time fast rates are shown to hold for combinatorial black-box optimization. As a second remark, we point out that obtaining these results is not really straightforward and requires novel work on bounding balls in combinatorial spaces as well as handling combinatorial structures (i.e. proofs of Lemma B3, B4, Prop B1, B4)
>
>
> 3. The choice of the tree is interesting. First, we point out that the choice of the ordering does not impact the theoretical results which only require the tree to satisfy Assumptions 3.1 and 3.3. However, to have a finer understanding of the impact of the ordering in practice, we performed the following additional  ablation study where we performed $10$ runs of OCTS with various trees. The results are reported below
>
> |               | Ising (20) |   CT (20)   |  LABS (20)  | MIS (20) |  Ising (50) | CT (50)     | LABS (50)   | MIS (50)    |
> |---------------|:----------:|:-----------:|:-----------:|:--------:|:-----------:|-------------|-------------|-------------|
> | $T + R$       |   20 (00)  |  4.00 (00)  | 7.33 (0.88) |  10 (00) |   50 (00)   |   10 (00)   | 5.17 (0.33) |   21 (1.0)  |
> | $T+R^*$       |   20 (00)  |  4.00 (00)  | 7.00 (0.88) |  10 (00) |   50 (00)   |   10 (00)   | 5.22 (0.46) |  21.2 (1.6) |
> | $\pi(T) + R$  |   20 (00)  | 3.80 (0.12) | 6.97 (0.72) |  10 (00) | 45.2 (2.71) | 8.76 (0.08) | 5.19 (0.22) |  19.2 (2.0) |
> | $\pi(T)+ R^*$ |   20 (00)  | 3.84 (0.32) |  6.24(0.62) |  10 (00) | 46.4 (1.49) | 8.55 (0.25) | 5.22 (0.32) | 20.6 (1.01) |
> | $\pi^*(T)+ R$ |   20 (00)  |  3.8 (0.0)  |  5.88(0.00) |  10 (00) |   50 (00)   |  9.40 (00)  |  4.32 (00)  |  21.2 (0.1)
>
> The table repots the best value observed after n=10*d**2 evaluations (with std) where $T$ denotes the tree of Figure 1 with $x_{l,i}=Bin_l(i) + \vec{0}_{d-l}$. $R$ denotes a root node sampled uniformly, $\pi(T)$ denotes a random permutation of the order of variables. Finally, R*  star set the root as the best points obtained from a RS with budget of $d$ evaluations. $\Pi^*(T)$ denotes the ordering where the variables in the tree are ranked according to the best function values recorded by switching the bit corresponding to the given variable and recording it ($d$ evaluations in total to get this ordering).  As it can be seen, on most test problems OCTS is robust to the choice of the root node in the sense that for a randomly chosen root (line T+R), the algorithm consistently finds similar optima with low std. Moreover, it is interesting to note that using random permutation ($\pi(T)+R$) does not improve the stability of the algorithm, which is due to the fact that on some problems (e.g. LABS and CT) there is a sequential link between the variables which is preserved by using $T$ and not when permuting the variables. Thus, in practice, it is recommended to keep the natural ordering of the variables. This is now included in the Appendix with a pointer in the main document.
>
> 4.  For the literature related to bandits, we honestly exactly had the same thoughts as the reviewer that there might be approaches in the bandit literature that are related to the current work. To the best of our knowledge, we surprisingly only found the works of [12, 31] to the present work. The fact that, somehow, the problem is deterministic changes a lot the nature of the algorithms where a lot of effort in the bandit literature is put to handle stochasticity.  However, if you have good references that are related to the work, we can definitely add them in the related work
>
> 5.   Thank you for pointing out that the empirical performance is a strong argument for the paper, which in our opinion validates the use for the Lipschitz approach of the paper. The details and hyperparameters of the baselines are all provided in Section D.1 of the Appendix (3 pages detailing the algorithms as well as hyperparameters). As far as we can see, all the hyperparameters as well as code for all the algorithms is provided
>
> 6. Thank you for pointing out the non-clarity of Algorithm 3 in Appendix. In order to fix this issue, we moved the image 5 (Appendix) which illustrates the algorithm in the document to make the connection with the OCTS more clear

---

> > ### Comment · Reviewer_n1bW · 2022-08-07
> > **Rebuttal**
> >
> > In the review I mentioned DIRECT as a similar strategy to handle the unknown Lipschitz constant case, and not really as an indication of incomplete related research.
> >
> > I do not have a clear pointer to which combinatorial bandit algorithm could be used directly, just that the problem structure is similar.
> >
> > I still have some reservations about the validity of Lipschitz condition in real problems, and I am not fully convinced about the ordering of the indices. The addtional experiments do indicate that this is not an issue, but intuitively I would find it surprising for a problem with heterogeneous input space.

---

> > > ### Author Response · Authors · 2022-08-09
> > > **Rebuttal**
> > >
> > > First, we appreciate that the reviewer took the time to read the rebuttal and we apologize for the misunderstanding about DIRECT.
> > >
> > > However, on top of the experiments and theory, it is hard to provide more evidence to debate the use of the Lipschitz constant/ordering. Nonetheless, we can give you our opinion/point of view on these topics in case that speaks to you.
> > >
> > > In our opinion, it is often the case that, in practice (like for the problems of the experiments we took from existing standardized benchmarks in various fields), the systems we wish to optimize are generally non-chaotic and have a good conditioning. Indeed, we generally wish to optimize systems that "represent" real systems and thus exhibit some smooth structure. Moreover, in applied settings, it is also often the case that the objective function of the problem is renormalized with regards to its various inputs before the optimization which further smoothen the system. It is generally an important aspect of the optimization pipeline. As a result, the systems we wish to optimize are often "smooth" in the sense that for similar inputs with large dimensions, they will have similar outputs $f(x) \approx f(x')$ for $x \approx x'$. In general, exploiting this smoothness signal can greatly help to optimize the system, e.g., captured in this work through the Lipschitz constant. In our opinion, this is what justified the large adoption and success of Bayesian methods (that use "smooth" kernels) and the DIRECT algorithm for continuous problems, although they might have limitations for instance over poorly conditioned systems (where most approaches might fail and a renormalization of the distance might make the trick). Here, we observe similar behaviors in the combinatorial world, which might even be stronger since its is known that all functions are Lipschitz as opposed to the continuous case. Finally, although it might not be the go-to solution for all problems, we point out that it is still important to note that this approach provides several results outperforming competitive benchmarks in various applications.
> > >
> > > Second, for the ordering of the index, it is possible that their effect/impact might be softened due to the large evaluation budget available when optimizing cheap-to-evaluate systems. Roughly speaking, if we have a budget of 100d**2 function evaluations for a given problem, due to the selection strategy of the OCTS, we know that the algorithm will reach the end of the tree around 100d times regardless the ordering (there are roughly d evaluations per round and each time a node at the end of the tree is evaluated). It means that regardless of the ordering, OCTS will go through all the variables (and check both possible combinations for each variable) at least 100d times which might strongly reduce the impact/variance of the ordering since 100d can be very large. However,  we agree that the results might be different for expensive-to-evaluate functions where we can only afford few evaluations (like in Bayesian optimization) and the ordering might have a stronger impact, which is however not the setup considered here.
> > >
> > > We hope that it provides some intuition to the reviewer.

---

> > > > ### Comment · Reviewer_n1bW · 2022-08-09
> > > > **Assumptions**
> > > >
> > > > In the case of contiuous systems the validity of Lipschitz condition is fairly strong. I am not as convinced for combinatorial problems, but probably there are enough such problems so that the algorithm can have wide enough applicability.
> > > >
> > > > I agree that many systems are sufficiently homogeneous, in which case the ordering is not a huge issue. Having a large budget also helps to alleviate the ordering issue, but as the authors pointed out, for a more limited budget the ordering could be a problem. Moreover, even if the overall budget is large, one would prefer an efficient use of the budget. As analogy, one can think of building decision trees, where the feature selection is crucial for the algorithms performance.

---

> > > > > ### Author Response · Authors · 2022-08-09
> > > > > **Assumptions**
> > > > >
> > > > > Thank you for pointing out the applicability of the approach.
> > > > > For the indices, we agree that the large budget (considered
> > > > > in the current setting) contributes  to this phenomenon.
> > > > > Even for decision tree, it might not be surprising that
> > > > > we observe a similar behavior by using a large number of samples.
> > > > > Similarly to random trees, we could also imagine to create
> > > > > on some problems a meta-strategy like random trees or similar techniques
> > > > > that use additional evaluations to further optimize the approach
> > > > > (which however seems to have a limited influence
> > > > > on the smooth problems of the paper in the large budget settings,
> > > > > but are now provided in the Appendix as an extension and cited in the core of the document).
> > > > >
> > > > > However, our view on this topic is the following:
> > > > > like decision trees, our approach is still a valuable tool
> > > > > and can even provide very good results without much further tuning
> > > > > on many problems, and we agree that it could always be further
> > > > > improved on some problems by fine-tuning the ordering or similarly
> > > > > designing budget dependent strategies or using different
> > > > > upper confidence bounds. In our opinion,
> > > > > it constitutes interesting extensions to
> > > > > the approach but does not impact the core (and dense)
> > > > > contribution which revolves about
> > > > > introducing a novel approach to solve black-box combinatorial
> > > > > problems. Indeed, it has to be recalled that
> > > > > the approach (1) provides
> > > > > state-of-the-art results on a wide variety of problems, (2) it introduces a novel approach to solve the combinatorial black-box problem
> > > > > and (3) we obtain relevant theoretical results as a byproduct.
> > > > > We are sincerely convinced (in our opinion) that it is still a good contribution
> > > > > for the optimization community.

---

### Official Review · Reviewer_S1XN · 2022-07-11

**Rating:** 7
**Confidence:** 2
**Soundness:** 3 good
**Presentation:** 4 excellent
**Contribution:** 3 good

**Summary:**

This paper presents an algorithm for solving combinatorial optimization problems where the objective function is a "black box" accessible only via an oracle. The algorithm is targeted at problems where this oracle is relatively cheap (as opposed to the standard Bayesian optimization setting), and is accompanied by finite time termination guarantees. The core algorithm relies heavily on Lipschitz constants to guide search and prune the tree; as this constant is often not known, the authors present a variant that instead only relies on the existence of a Lipschitz constant. The authors conclude with a computational analysis of the performance of the algorithms as a function of the number of function evaluations.

**Questions:**

* Please consider moving the qualification as to why Bayesian optimization methods are not included in the computational study from the Appendix to Section 6. I am also not sure how convincing I find the author's argument on this point: even if Bayesian optimization is much slower per iteration, if it nonetheless provides good solutions after relatively few function evaluations then this provides an interesting baseline for comparison with the new algorithms.
* p22 l756: There is a typo in at least one of the convergence rates (I think the first).

**Limitations:**

There is no explicit discussion of potential negative societal impact.

**Strengths And Weaknesses:**

The paper: presents a novel algorithm in an area of interest to the NeurIPS community, includes interesting theoretical results, and is clearly written. The only weakness I can identify is the lack of a computational comparison against Bayesian optimization techniques (see "Questions").

---

> ### Author Response · Authors · 2022-08-02
> **Official Response to Reviewer S1XN**
>
> First, we would like to thank the reviewer for its feedback on the paper and the general positive review.  We thank you for noticing that the paper aims at optimizing functions with cheap-to-evaluate cost. We try to address the main questions below:
>
> 1. We directly move the qualification as to why Bayesian optimization methods are not included in the computational study directly in Section 6. Moreover, in order to make it more clear since the beginning, we now stress in the introduction that we aim at providing algorithms that are tailored to optimize function with cheap-to-evaluate cost. Recall that the Bayesian method are not compared to the current methods, as their heavy computational cost to generate the next evaluation point does not make them suitable for cheap-to-evaluate black-boxes. In order to have a better idea, we computed the following time to sample the next evaluation points:
>
> |                                             | OCTS   | GA     | EA             | RS     | RLS    | GHC    | SA     | Bayesian                                                       |
> |---------------------------------------------|--------|--------|----------------|--------|--------|--------|--------|----------------------------------------------------------------|
> | Complexity to sample $x_{t+1}$              | $O(d)$ | $O(\lambda)$ | $O(1)$         | $O(1)$ | $O(1)$ | $O(1)$ | $O(1)$ | Solving a BB problem of dim $d$  ($O(2^d)$ for an exact solution)  |
> | Memory to compute $x_{t+1}$                 | $t+1$  | $\lambda (30)+1$  | $\lambda (30)$ | $1$    | $2$    | $2$    | $2$    | $t+1$                                                          |
> | Time to compute $x_{t+1}$ after $t=100$ | 0.001 (s)| 0.004 (s) | 0.0009 (s)         | 0.0007 (s) | 0.0008 (s) | 0.0008 (s) | 0.0007 (s) | 62.00 (s)
>
> We took COMBO [41] for the Bayesian method as well as their official implementation. The time to compute $x_{t+1}$ is measured on the contamination problem (d=25) on a i7 CPU @ 1.80GHz   1.99 GHz with 16GB of RAM.  As it can be seen, OCTS is in the same order of magnitude as other methods (milliseconds). On the other side, Bayesian methods literally take 1 minute to query a novel point. More precisely, since it takes ~1 hour for the current Bayesian method to perform 100 function evaluations, it might be even faster to perform an exhaustive search that querying a single point in some cases (where the function is cheap to evaluate).  However, to have an idea of a simple comparison of the proposed algorithms with Bayesian optimization, we performed the following experiment: we took the algorithms of the paper with the budget of the experiments set to 100*d^2 evaluations (as set in the paper) and compared it to Bayesian optimization with a budget of 100 evaluations.
>
> |                | OCTS         | GA           | EA           | RS           | RLS          | GHC          | SA            | Bayesian        |
> |----------------|--------------|--------------|--------------|--------------|--------------|--------------|---------------|-----------------|
> | Number of eval | 62500        | 62500        | 62500        | 62500        | 62500        | 62500        | 62500         | 100             |
> | Contamination  0.0 | -21.35 (32s) | -21.42 (43s) | -21.35 (28s) | -21.57 (24s) | -21.52 (24s) | -21.61 (21s) | -21.43 (23s)  | -21.57 (88min)  |
> | Contamination 0.01  | -21.52 (34s) | -21.60 (45s) | -21.56 (30s) | -21.73 (23s) | -21.68 (22s) | -21.78 (22s) | -21.61 (24s) | -21.74 (91min)  |
> | Contamination 0.0001 | -21.35 (35s) | -21.45 (45s) | -21.36 (25s) | -21.57 (23s)    | -21.49 (23s)      | -21.61    (21s)   | -21.45    (23s)    | -21.72 (87min)
>
> As it can be seen, for cheap-to-evaluate black-box system, we can only query 100 points in more than an hour with Bayesian methods while with other methods we can easily query 50000 points. Of course, it results in a less competitive algorithm with such a different budget (higher is better). However, they are suitable for problems where the black-box evaluation is generally significantly larger than the time to generate the next sample point, and we strongly advise using them in this case.
>
> 2. Thank you for the typo in the convergence rate
>
> 3. We apologize if the section about the potential negative societal impact. It is now in the document and here is an extract:
>
> In our work, we proposed a novel methodology to optimize binary functions with cheap-to-evaluate cost. These new solvers are mostly agnostic to the specific application, and can be applied in a wide range of optimization problem (ranging from graph analysis, to electronic design). Therefore, the societal and ethical impacts of our contribution are heavily dependent on the nature of the problems solved with the algorithm. We start by noting that beneficial applications of OCTS are thick on the ground, ranging from the design more efficient telecommunication applications, to the control of contaminations.

---

### Official Review · Reviewer_3NSu · 2022-07-12

**Rating:** 7
**Confidence:** 3
**Soundness:** 4 excellent
**Presentation:** 3 good
**Contribution:** 4 excellent

**Summary:**

The paper proposes two novel methods for combinatorial black-box optimization (i.e. over an unconstrained binary domain) based on optimistic tree search, one based on a known Lipschitz constant (OLTS) and another one when it is unknown (OCTS). The general idea of the OLTS is to evaluate nodes in a tree with large upper bounds in their subtrees, where the upper bound is based on the Lipschitz constant and the diameter of the subtree. This is extended in OCTS when the Lipschitz constant is not known by searching a superset of nodes that would contain the node in OLTS. Both methods are proven to have linear convergence rates (with a dependence on the Lipschitz constant). Computational experiments show that OCTS outperform several other heuristic-based methods.

**Questions:**

These are only presentation issues, but they nevertheless should be addressed.

1. The introduction claims that these methods are "provable", which is unfortunately not discussed further. A natural interpretation might be that the algorithm finds the provably optimal solution for some reasonable budget. However, at least OCTS does not return a provably optimal solution without exhaustively searching all of the $2^d$ points. On the other hand, if one would include exhaustive search in the definition of "provable", then this is not a meaningful claim. If there is no reasonable explanation for what "provable" means in this context (and I believe there is not), this qualification only adds confusion and should not be used.

2. Could you include some discussion about solve/iteration time compared to the baselines, even if briefly? Preferably, if you have computational results on this, that would be best. My impression is that the iterations should be comparable to the baselines in the proposed method, but as is I cannot exactly verify if OCTS is indeed faster than other methods. This might be more suitable for the appendix but I would add a sentence referring to it in the main text.

3. The paper defines the algorithms relative to a class of tree structures. However, the computational section does not specify exactly which tree was used. Could you mention that in the paper? In addition, I would have liked a brief discussion on the effect of using different trees (e.g. does the tree really make a difference?), but this is not essential.

4. I would adjust the statement in p.9: "no efficient algorithm exists to find maximum independent sets". Although the problem is NP-hard, this is not accurate if this is interpreted as "efficient in practice". It is fine to say that there is no known polynomial-time algorithm (although this is already implied by the NP-hardness), but maximum independent set is reasonably tractable in practice.

5. In MaxSAT in p.9, please clarify that the "optimum that no other algorithm could find" is with respect to the baselines examined in the paper. Since the previous sentence talks about specialized solvers, this clarification would help avoid the misinterpretation that this statement includes them as well.

6. I am aware that this is mentioned in Section 2.1, but I suggest mentioning in the introduction that this method is tailored to black-box problems where evaluation is not expensive. This is so that, first, readers can quickly identify if this is a good approach for their black-box problem, and second, you immediately establish the scope of your method, i.e. you are not comparing against methods that may have expensive iterates such as model-based methods. If you have space, I would even briefly reiterate this when defining the baselines in Section 6.2.

7. Please run a spell checker for the camera-ready version, there are many typos in the paper. Here are some of them: l.54, "dimensionalitites"; l.154, "a-priory"; l.163, $x_{l,i}$ and $\mathcal{X}_{l,i}$ should be indexed by $l_t$ and $i_t$; l.195, "Moroever"; l.202, repeated "order", l.264: "bt", l.265: "iteration" should be plural; l.289: "dimensionailites"; l.318, "runing"; l.570: "maximimum"; l.782: "Real-word problems"; [18] and [19] are the same reference.

**Limitations:**

No limitations besides the ones discussed above.

**Strengths And Weaknesses:**

The black-box methods proposed in the paper are very appealing: they are simple to implement, theoretically grounded, and appear to work well in practice. The approach appears to be original as far as I am aware. The computational section is sufficiently extensive, with six different problem classes and one experiment to illustrate the convergence rates, and the method generally outperforms the baselines. I particularly appreciate the theoretical guarantees and their computational analysis in Section 6.1.  The paper could have benefited from a comparison with model-based methods, but I believe it is not too unreasonable to omit them given that they typically have more expensive iterations. The presentation is overall clear, but there are several minor issues that need to be addressed below.

Most of my comments below are regarding presentation, which should be fixable. Assuming those are addressed, I recommend acceptance for this paper.

---

> ### Author Response · Authors · 2022-08-02
> **Official Response to Reviewer 3NSu**
>
> First, we would like to thank the reviewer for their feedback. We try to provide an answer to the questions below:
>
> 1. We agree with the reviewer that the word "provable" is indeed not appropriate in the context of black-box optimization. More precisely, we agree that only the algorithms that search all the $2^d$ points could be said to be provable in this setting. To avoid this confusion, we now use the term "with provable guarantees" in the paper.
>
> 2. The solve/iteration time question is interesting. Since this time heavily depends on the implementation details, hardware and tricks; we only measured the solve time in the paper by recording the number of black-box calls. However, to provide some details about the computational time, we added the following experiments in the appendix.
>
> |                                             | OCTS   | GA     | EA             | RS     | RLS    | GHC    | SA     | Bayesian                                                       |
> |---------------------------------------------|--------|--------|----------------|--------|--------|--------|--------|----------------------------------------------------------------|
> | Complexity to sample $x_{t+1}$              | $O(d)$ | $O(\lambda)$ | $O(1)$         | $O(1)$ | $O(1)$ | $O(1)$ | $O(1)$ | Solving a BB problem of dim $d$  ($O(2^d)$ for an exact solution)  |
> | Memory to compute $x_{t+1}$                 | $t+1$  | $\lambda (30)+1$  | $\lambda (30)$ | $1$    | $2$    | $2$    | $2$    | $t+1$                                                          |
> | Time to compute $x_{t+1}$ after $t=100$ | 0.001 (s)| 0.004 (s) | 0.0009 (s)         | 0.0007 (s) | 0.0008 (s) | 0.0008 (s) | 0.0007 (s) | 62.00 (s)                                                         |
>
> The time to compute $x_{t+1}$ is measured on the contamination problem (d=25) on a i7 CPU @ 1.80GHz   1.99 GHz with 16GB of RAM. As it can be seen, OCTS is in the same order of magnitude as other methods and significantly faster than Bayesian methods.
>
> 3. The choice of the tree was initially specified in line 793 of the appendix (e.g. the Tree of Section 3 with random initial point), but it is now detailed directly in Section 6. Moreover, to have a better understanding of the impact in practice, we performed (ten) additional runs of  OCTS  using the various trees reported below:
>
> |               | Ising (20) |   CT (20)   |  LABS (20)  | MIS (20) |  Ising (50) | CT (50)     | LABS (50)   | MIS (50)    |
> |---------------|:----------:|:-----------:|:-----------:|:--------:|:-----------:|-------------|-------------|-------------|
> | $T + R$       |   20 (00)  |  4.00 (00)  | 7.33 (0.88) |  10 (00) |   50 (00)   |   10 (00)   | 5.17 (0.33) |   21 (1.0)  |
> | $T+R^*$       |   20 (00)  |  4.00 (00)  | 7.00 (0.88) |  10 (00) |   50 (00)   |   10 (00)   | 5.22 (0.46) |  21.2 (1.6) |
> | $\pi(T) + R$  |   20 (00)  | 3.80 (0.12) | 6.97 (0.72) |  10 (00) | 45.2 (2.71) | 8.76 (0.08) | 5.19 (0.22) |  19.2 (2.0) |
> | $\pi(T)+ R^*$ |   20 (00)  | 3.84 (0.32) |  6.24(0.62) |  10 (00) | 46.4 (1.49) | 8.55 (0.25) | 5.22 (0.32) | 20.6 (1.01) |
> | $\pi^*(T)+ R$ |   20 (00)  |  3.8 (0.0)  |  5.88(0.00) |  10 (00) |   50 (00)   |  9.40 (00)  |  4.32 (00)  |  21.2 (0.1) |
>
>
> where $T$ denotes the tree of Figure 1 with $x_{l,i}=Bin_l(i) + \vec{0}_{d-l}$, $R$ denotes a root node sampled uniformly, $\pi(T)$ denotes a random permutation of the order of variables. Finally, $R^*$ set the root as the best points obtained from a RS with a budget of $d$ evaluations. $\Pi^*(T)$ denotes the ordering where the variables in the tree are ranked according to the best function values recorded by switching the bit corresponding to the given variable and recording it ($d$ evaluations in total to get this ordering).
> As it can be seen, on most test problems OCTS is robust to the choice of the root node in the sense that for a randomly chosen root (line T+R), the algorithm consistently finds similar optima with low std. Moreover, it is interesting to note that using random permutation ($\pi(T)+R$) does not improve the stability of the algorithm, which is because on some problems (e.g. LABS and CT) there is a sequential link between the variables which is preserved by using $T$ and not when permuting the variables. Thus, it is recommended to keep the natural ordering of the variables. This is now included in the Appendix with a pointer in the main document.
>
> 4 - 5 - 6 - 7. We thank the reviewer for pointing out the confusing statement on p.9: "no efficient algorithm exists to find maximum independent sets". It is now replaced with "there is no known polynomial-time algorithm" in the final document. Similarly, for MaxSAT in p.9, it is now written: "optimum that no other algorithm could find by other baselines". Finally, we added both in the introduction and experimental section that our goal is to design algorithms that aim to optimize cheap-to-evaluate functions. All the typos are now corrected.

---

> > ### Comment · Reviewer_3NSu · 2022-08-09
> > **Response to authors**
> >
> > Thank you for the thorough response. The table with iteration times is a valuable addition to the paper. I continue to recommend acceptance.

---

> > > ### Author Response · Authors · 2022-08-09
> > > **Response to reviewer**
> > >
> > > Finally, we would like to thank the reviewer for taking the time to read the rebuttal and we are happy that it helped to clarify some of its questions.

---

### Meta-Review · Area_Chair_yuU6 · 2022-08-26

**Recommendation:** Accept
**Confidence:** Less certain

**Metareview:**

This paper proposes two methods for black box optimization of Lipschitz combinatorial binary functions. The reviewers agree that the paper is well written, the methods are sufficiently novel, and that the results are of interest to the NeurIPS community. The main drawback with the paper is that reviewer n1bW felt that the theoretical results are straightforward (but nevertheless useful). Several reviewers also had hoped for comparisons with Bayesian optimization techniques, but during the discussion period it was decided that this comparison can be omitted due to the much higher computational cost of Bayesian methods. I tend to agree with the reviewers that this paper is above the bar for NeurIPS.

**Award:**

No

---

### Decision · Program_Chairs · 2022-09-14

Accept